



# The Importance of Size Ranges in Aerosol Instrument Intercomparisons: A Case Study for the ATom Mission

Hongyu Guo[1,2], Pedro Campuzano-Jost[1,2], Benjamin A. Nault[1,2], Douglas A. Day[1,2], Jason C. Schroder[1,2,*], Jack E. Dibb[3], Maximilian Dollner[4], Bernadett Weinzierl[4], and Jose L. Jimenez[1,2]

[1]Department of Chemistry, University of Colorado Boulder, Boulder, CO, 80309, USA
[2]Cooperative Institute for Research in Environmental Sciences, University of Colorado Boulder, Boulder, CO, 80309, USA
[3]Earth Systems Research Center, Institute for the Study of Earth, Oceans, and Space, Univ. of New Hampshire, Durham, NH, 03824, USA
[4]Faculty of Physics, University of Vienna, Vienna, Austria
[*]Now at Air Pollution Control Division, Colorado Department of Public Health and Environment, Denver, CO, 80246, USA
*Correspondence to*: Jose L. Jimenez (jose.jimenez@colorado.edu)

**Abstract.** Aerosol intercomparisons are inherently complex, as they convolve instrument-dependent detection efficiencies vs. size (which often change with pressure, temperature, or humidity) and variations on the sampled aerosol population, in addition to differences in chemical detection principles (e.g., including inorganic-only nitrate vs. inorganic plus organic nitrate for two instruments). The NASA Atmospheric Tomography Mission (ATom) spanned four separate aircraft deployments, which sampled the remote marine troposphere from 86°S to 82°N over different seasons with a wide range of aerosol concentrations and compositions. Aerosols were quantified with a set of carefully characterized and calibrated instruments, some based on particle sizing and some on composition measurements. This study aims to provide a critical evaluation of the size-related factors impacting aerosol intercomparisons, and of aerosol quantification during ATom, with a focus on the Aerosol Mass Spectrometer (AMS). The volume determined from physical sizing instruments is compared in detail with that derived from the chemical measurements of the AMS and the Single Particle Soot Photometer (SP2). Special attention was paid to characterize the upper end of the AMS size-dependent transmission with in-field calibrations, which we show to be critical for accurate comparisons across instruments with inevitably different size cuts. Observed differences between campaigns emphasize the importance of characterizing AMS transmission for each instrument and field study for meaningful interpretation of instrument comparisons. Good agreement was found between the composition-based volume (including AMS-quantified sea salt) and that derived from the size spectrometers. The very clean conditions during most of ATom resulted in substantial statistical noise (i.e., precision error), which we show to be substantially reduced by averaging at several-minute time intervals. The AMS captured, on average, $95 \pm 15\%$ of the standard $PM_1$ volume. These results support the absence of significant unknown biases and the appropriateness of the accuracy estimates for AMS total mass/volume for the mostly aged air masses encountered in ATom. The particle size ranges that contribute chemical composition information to the AMS and complementary composition instruments are investigated, to inform their use in future studies.





## 1 Introduction

Aerosols are ubiquitous in the atmosphere and have a lifetime of about a week, and thus can travel long distances (Tsigaridis et al., 2014), and have important effects on climate forcing, through both direct (Pilinis et al., 1995; Haywood and Boucher, 2000) and indirect effects (Lohmann and Feichter, 2005; IPCC, 2013). Remote regions account for much of the Earth's surface and are infrequently sampled, and thus have especially uncertain aerosol distributions and radiative impacts (IPCC, 2013; Hodzic et al., 2020). The NASA Atmospheric Tomography Mission (ATom) sampled the remote marine troposphere from 86°S to 82°N over four different seasons with a comprehensive suite of high-quality and carefully calibrated and operated physical and chemical aerosol instruments. It provides a unique dataset to improve our understanding of the remote atmospheric aerosols and thus refine global model predictions. A prerequisite for that purpose is to evaluate the accuracy and consistency of the ATom aerosol instruments.

The ATom physical sizing instruments have been recently described and evaluated in Williamson et al. (2018), Kupc et al. (2018), and Brock et al. (2019), while the Particle Analysis by Laser Mass Spectrometer (PALMS) chemical instrument during ATom has been described in Froyd et al. (2019). In this paper, we focus on the Aerodyne Aerosol Mass Spectrometer (AMS). AMS (Canagaratna et al., 2007) and Aerosol Chemical Speciation Monitor (ACSM, smaller, lower cost, and simpler to operate versions) (Ng et al., 2011), have been deployed extensively worldwide for ground aerosol monitoring (Jimenez et al., 2009; Crenn et al., 2015; Hu et al., 2015; Kiendler-Scharr et al., 2016; Zhang et al., 2018; ACTRiS, 2019). AMS has been deployed in most advanced atmospheric chemistry aircraft experiments worldwide (Dunlea et al., 2009; Middlebrook et al., 2012; Barth et al., 2015; Schroder et al., 2018; Garofalo et al., 2019; Hodzic et al., 2020; Mei et al., 2020; Morgan et al., 2020). The overall AMS concentration uncertainty ($2\sigma$) is normally reported as ±38% for organic aerosol (OA) and ±34% for inorganics, while the precision is typically much better, except at concentrations near the detection limit (Bahreini et al., 2009; Jimenez et al., 2016). A detailed evaluation of those uncertainties requires both very careful AMS characterization and calibration, as well as high-quality collocated measurements, as was the case in ATom.

This work uses the extensive ATom field dataset for remote aerosols to evaluate (1) the consistency of the different submicron aerosol volume measurements, (2) the quantification ability of the





AMS for remote aerosols, and (3) the size ranges contributing chemical composition information to
different instruments for ATom, and their variation with altitude. Volume comparisons probe the ability
of the AMS to quantify total mass and predict aerosol density based on fractional composition accurately,
and hence is the most germane comparison for total quantification. We examine in detail the accurate
quantification and application of the AMS transmission efficiency ($E_L$) to the particle volume
intercomparisons in this study. This study also serves as the basis for a future study on individual chemical
species intercomparisons.
**2 Methods**
**2.1 ATom overview**

The Atmospheric Tomography Mission (ATom) consisted of four series of flights onboard the

NASA DC-8 over the middle of the Pacific and Atlantic oceans, spanning from 82° N to 86° S latitude,
to characterize the composition and chemistry of the global background troposphere. Over two years, the
DC-8 aircraft was deployed once a season: July-August 2016 (ATom-1), January-February 2017 (ATom-
2), September-October 2017 (ATom-3), and April-May 2018 (ATom-4). During these flights, the DC-8
repeatedly ascended and descended between ~0.18 and ~13 km altitudes at regular intervals, typically
every hour (with a single vertical profile lasting ~25 min), leading to executing ~140 vertical profiles of
the troposphere per deployment. The unique spatio-temporal coverage and high-quality measurements of
this campaign ensure that its data will be used very widely, such as to evaluate and constrain global
modeling. Therefore it is of high interest to document the consistency of the multiple aerosol
measurements. This analysis is also useful to re-evaluate the quantification uncertainties of the AMS for
a wide range of particle concentrations and composition (e.g., Fig. S1 in the supplementary info, SI). Due
to the similarities in the geographic coverage of ATom studies, we focus on the intercomparisons for the
first two ATom campaigns in the following analysis.
**2.2 Definitions of particle diameters**

Conversions between different particle diameter definitions are required for meaningful

instrument comparisons. For example, particle size spectrometers report estimated geometric diameter





$(d_p)$, which is derived from multiple condensation particle counters using an inversion method, or from light scattering signals by using an assumed constant refractive index for aerosols. AMS transmission operates in vacuum aerodynamic diameter $(d_{va})$ since its aerodynamic lens and supersonic expansion operate in the free molecular regime (DeCarlo et al., 2004). Impactors (Marple et al., 1991, 2014) and cyclones (typically sourced from URG Corp., Chapel Hill, NC, USA) are often installed upstream of aerosol instruments to preselect desired aerosol ranges for ground or aircraft measurements. The cutoff sizes of both devices follow the transition-regime aerodynamic diameter $(d_{ta}$; as the size range of interest to this study is in the transition regime, requiring a "slip correction"). A detailed discussion of particle diameters definitions can be found in DeCarlo et al. (2004). $d_{va}$ is related to the volume-equivalent diameter $(d_{ve}$, the diameter that would result if the particle was melted to form a sphere of the same density as the particle and without any internal voids) as:

$$d_{va} = \frac{\rho_p}{\rho_0} \frac{d_{ve}}{\chi_v} \qquad (1)$$

where $\rho_p$ is the particle density, $\rho_0$ is the standard density (1 g cm$^{-3}$), and $\chi_v$ is the vacuum (i.e., free-molecular regime) dynamic shape factor (=1 for spheres and >1 for non-spherical particles). Since the aerosols sampled during ATom were remote and aged, we assume $\chi_v \sim 1$ and $d_{ve} \sim d_p$. The transition-regime aerodynamic diameter can be calculated as:

$$d_{ta} = d_{ve} \sqrt{\frac{1}{\chi_t} \frac{\rho_p}{\rho_0} \frac{C_c(d_{ve})}{C_c(d_{ta})}} \qquad (2)$$

where $\chi_t$ is the transition-regime dynamic shape factor, and $C_c$ is the Cunningham slip correction factor. In this study, $\chi_t$ is assumed as 1 and $C_c$ is calculated based on air pressure. Although a given particle always has the same dry $d_p$ and $d_{va}$, the dry $d_{ta}$ changes with pressure. To distinguish the $d_{ta}$ calculated at different altitudes, we use $d_{ta,sea}$ to denote that calculated at sea level ($P$ = 1013 mbar) and $d_{ta,air}$ for sampling aloft with an aircraft (or at an elevated ground site). In addition, all diameters change under humid/dry conditions due to water uptake or evaporation (DeCarlo et al., 2004).





## 2.3 AMS description and quantification

The highly customized University of Colorado (CU) high-resolution time-of-flight aerosol mass spectrometer (HR-ToF-AMS, hereafter referred to as AMS; Aerodyne Research Inc., Billerica, MA) (DeCarlo et al., 2006) measured non-refractory, bulk submicron particles composition at 1 Hz resolution. The AMS uses an aerodynamic lens to sample particles into a high vacuum, where they impact and vaporize on a hot porous tungsten vaporizer (600 °C). The evaporated constituents undergo electron ionization (EI), with the resulting ions being detected by a mass spectrometer (Jayne et al., 2000; Jimenez et al., 2003; Drewnick et al., 2005; DeCarlo et al., 2006; Canagaratna et al., 2007). The mass concentration of a species, $s$, within a multi-component aerosol particle can be calculated from the measured ion signal with the following equation (Alfarra et al., 2004; Canagaratna et al., 2007; Jimenez et al., 2016):

$$C_s = \frac{10^{12}}{CE_s} \frac{MW_{NO3}}{RIE_s IE_{NO3} Q N_A} \sum_{all\ i} I_{s,i} \qquad (3)$$

where $C_s$ is the mass concentration of species $s$, $MW_{NO3}$ is the molecular weight of nitrate, $CE_s$ is the collection efficiency of species $s$, $RIE_s$ is the relative ionization efficiency of species $s$ (to nitrate), $IE_{NO3}$ is the ionization efficiency of nitrate, $Q$ is the volume flow rate into the AMS, $N_A$ is Avogadro's number, $I_{s,i}$ is the ion signal from ion $i$ produced from species $s$, and the $10^{12}$ factor accounts for unit conversions.

$CE$ is typically defined as the efficiency with which particles entering the AMS inlet are detected. It has been formally defined as a product of aerodynamic lens transmission efficiency for spherical particles ($E_L$), transmission efficiency correction for non-spherical particles ($E_s$) due to additional particle beam broadening, and detection efficiency at the vaporizer ($E_b$), which can be reduced due to particle bounce. It is thus expressed as

$$CE = E_L \times E_s \times E_b \qquad (4)$$

(Huffman et al., 2005; Canagaratna et al., 2007; Middlebrook et al., 2012). Previous studies have shown that $E_s \sim 1$ for ambient particles (Huffman et al., 2005; Salcedo et al., 2007), and thus $CE$ is determined by $E_L$ and $E_b$. When the mass size distribution being sampled is mostly within the region where $E_L \sim 1$, then $CE \sim E_b$. Most literature papers make that implicit approximation, although it is not clear that the approximation is always justified, since $E_L$ changes in time and between instruments and is infrequently





quantified as it is experimentally challenging to do so. $E_b$ depends on particle viscosity and thus phase
(Matthew et al., 2008; Middlebrook et al., 2012; Pajunoja et al., 2016). With the "standard vaporizer"
used in this study (Hu et al., 2020), ambient aerosols in continental regions typically have $E_b$~0.5, but a
range between 0.5 to 1 can be observed (Middlebrook et al., 2012; Hu et al., 2017, 2020). $E_b$ increases
for certain compositions that lead to less viscous particles, such as high ammonium nitrate mass fraction
or high acidity conditions, which can be estimated with a parameterization based on aerosol composition
(Middlebrook et al., 2012; Hu et al., 2017, 2020; Nault et al., 2018). Such parametrizations assume
internally mixed aerosols, which is typically the case for submicron ambient aerosol away from sources
due to condensation and coagulation (Petters et al., 2006; Wang et al., 2010; Mei et al., 2013). $CE$ is
estimated to contribute substantially to the overall uncertainty of AMS concentration measurements
(Bahreini et al., 2009).
The main submicron inorganic ambient aerosol species are ammonium ($NH_4$), sulfate ($SO_4$),
nitrate ($pNO_3$), and chloride (Chl), and in marine areas, sea salt. The charges are omitted for the AMS-
measured nominally inorganic species, as the AMS may also detect some $SO_4$ or $NO_3$ signals from
organosulfates or organonitrates (Farmer et al., 2010). To avoid the confusion between the $NO_3$ radical
and particle $NO_3$, $pNO_3$ is used to denote total particle $NO_3$ explicitly (Nault et al., 2018). $RIEs$ for the
inorganic species can be calibrated regularly (including in the field). However, similar explicit
calibrations cannot be readily performed for the thousands of individual organic aerosol (OA) molecules
in ambient particles. Thus, laboratory-based calibrations with a limited set of OA species have been used
to estimate $RIE_{OA}$ (Slowik et al., 2004; Dzepina et al., 2007; Jimenez et al., 2016; Robinson et al., 2017;
Xu et al., 2018), and this approach has been verified using laboratory and field intercomparisons with
other instruments (Takegawa et al., 2005; Dzepina et al., 2007; DeCarlo et al., 2008; Bahreini et al., 2009;
Dunlea et al., 2009; Timonen et al., 2010; Docherty et al., 2011; Middlebrook et al., 2012; Crenn et al.,
2015). Bahreini et al. (2009) estimated the uncertainty in $RIE_{NH4}$ (which is always calibrated in the field)
to be ~10% vs. 15% for the other inorganics (sulfate, chloride; since most AMS users do not perform in-
field calibrations for those or do so less frequently). Compared to the inorganics, the uncertainty in $RIE_{OA}$
was estimated to be higher at 20%, to account for the diversity of species (Bahreini et al., 2009). An
average $RIE_{OA}$~1.4 was determined from laboratory calibrations. However, there are conflicting reports





for $RIE_{OA}$ of chemically-reduced species such as hydrocarbons, with some values around 1.4 and others
higher (Slowik et al., 2004; Dzepina et al., 2007; Docherty et al., 2011; Jimenez et al., 2016; Reyes-
Villegas et al., 2018; Xu et al., 2018). However, such species were insignificant during ATom. For more
oxidized species, relevant to most biomass burning OA and secondary organic aerosol (SOA), average
laboratory $RIE_{OA}$ overlaps within uncertainties of 1.4 (Jimenez et al., 2016; Xu et al., 2018). Reviews on
this topic (Jimenez et al., 2016; Murphy, 2016a, 2016b) have emphasized the need for additional
investigation of AMS quantification in the field. For the AMS reported mass concentration, uncertainties
(i.e., accuracies) in $CE$ (30%), $RIE$s (20% for OA, 15% for SO$_4$/Chl, and 10% for NH$_4$), and $IE_{NO3}$ (10%)
dominate the total reported uncertainties in most situations, although precision (statistical) error becomes
important at low concentrations and short averaging times.

**2.4 AMS operation during ATom**

The aircraft operation of the CU AMS has been discussed previously (DeCarlo et al., 2006, 2008,
2010; Dunlea et al., 2009; Cubison et al., 2011; Kimmel et al., 2011; Schroder et al., 2018). The specific
operational procedures used during ATom have been discussed in Nault et al. (2018) and Hodzic et al.
(2020). Important operation details of AMS that are relevant to this study are described below. Per aircraft
conventions, mass concentrations are reported at μg sm$^{-3}$ (microgram per cubic meter air volume at
standard conditions of $T$ = 273.15 K and $P$ = 1013 mbar, hereafter referred to as STP. Note that many
definitions of STP are in use, especially in other fields).
Ambient aerosols were sampled through an NCAR High-Performance Instrumented Airborne
Platform for Environmental Research (HIAPER) Modular Inlet (HIMIL) (Stith et al., 2009) mounted on
a 4" raised platform on the window plate to ensure that sampling occurred consistently outside the DC-8
boundary layer (Vay et al., 2003). Aerosols were introduced at a constant standard flow rate of 9 sL min$^{-1}$
(up to ~9 km, 15 L min$^{-1}$ above that; "s" refers to standard conditions, and no "s" indicates a volumetric
flow at in-situ $T$ and $P$), with 1 L min$^{-1}$ being continuously subsampled into a pressure controlled inlet
(PCI) operated at 250 mbar (187 Torr) (Bahreini et al., 2008). A fraction of that flow, 94 scm$^3$ min$^{-1}$, was
then sampled into the high vacuum region of the mass spectrometer through an aerodynamic focusing
lens operated at 2.00 mbar (1.50 Torr). Due to the much lower ambient air pressure at high altitudes, the



PCI pressure cannot be maintained at 250 mbar above ~9 km, resulting in a drop in lens pressure (down
to 1.00 Torr) and flow (down to 55 scm$^3$ min$^{-1}$) at the max altitude (12.5 km). Residence times from the
tip of the HIMIL to the aerosol vaporizer varied from ~0.4 s in the boundary layer and ~0.9 s at 12 km
during ATom (Fig. S2 in SI; note that, a detailed characterization of HIMIL and PCI performance is
included in SI as Sect. 4 with Figs. S2-S9). The relative humidity (RH) in the line was not actively
controlled but was very low, on average $10 \pm 21$ % in ATom-1 and -2 with a median of 0.4%, due to the
thermal gradients between the plane cabin and ambient ($T_{rack}$ - $T_{ambient}$ = $27 \pm 13$ K) (8% of the data
was >40% RH, including 3% >80% RH, which could increase $CE$). Composition-dependent $CE$ was
estimated based on the Middlebrook et al. (2012) parameterization and was on average $0.87 \pm 0.15$ and
$0.90 \pm 0.13$ for ATom-1 and -2, respectively, mainly due to high acidity (Fig. S10). After every research
flight, $IE_{NO3}$ was calibrated by atomizing pure $NH_4NO_3$ solutions and selecting 400 nm (mobility
diameter, $d_m$; equivalent to $d_{va}$ = 550 nm) (DeCarlo et al., 2004) particles with a differential mobility
analyzer (DMA, TSI model 3081, St. Paul, MN, USA) into AMS. $RIE$s for sulfate, ammonium, and
chloride were determined by multiple in-field calibrations.
A summary plot of the in-field calibrations of these parameters is shown as Fig. S11. Assuming a
constant instrument response over the course of each deployment, the variability of the calibrations can
be taken as an estimate of the random component of $RIE$ uncertainty. Uncertainties ($2\sigma$) for $RIE_{NH4}$,
$RIE_{SO4}$, and $RIE_{Chl}$ are hence 4% (6%), 4% (2%), and 5% (8%), respectively for ATom-1 (ATom-2), all
smaller than the reported values from Bahreini et al. (2009). The $2\sigma$ variability of $IE_{NO3}$ (normalized as
its ratio to the air beam signal, $IE_{NO3}/AB$) is 6% for ATom-1 and 15% for ATom-2. The propagated AMS
uncertainties using these values, 31% for inorganics and 37% for organics, are similar to those from
Bahreini et al. (2009), due to the dominant uncertainty contribution from $CE$ (30%).
$IE_{NO3}$ calibrations, performed in event trigger mode with 400 nm ammonium nitrate aerosol
(Schroder et al., 2018), also provided multiple AMS transmission measurements throughout the
campaign, by a direct comparison of the single-particle AMS counts with a Condensation Particle Counter
(CPC) (Nault et al., 2018). Besides this single-size (at the edge of the $E_L$~1 range) post-flight calibrations,
the upper end of the AMS transmission curve was characterized on the aircraft during ATom-2 by
measuring multiple sizes of monodisperse ammonium nitrate ($d_m$ range 350-850 nm). The resulting





transmission accounts for all the losses in the PCI and aerodynamic lens. A calculation of the inlet line losses is presented in the SI, and based on these calculations additional losses are very small and can be ignored. These calculations do not include the transmission of the actual HIMIL aircraft inlet (Stith et al., 2009), nor the secondary diffuser inside the HIMIL. To confirm the aircraft probe related size-dependent losses or enhancements did not impact the overall transmission, the AMS sampled several times at different altitudes off the University of Hawaii (UH)/NASA Langley Aerosol Research Group (LARGE) inlet used by the NOAA instruments over the course of the four ATom deployments, which transmits particles to ~3-5 µm $d_{ta,air}$ with 50% passing efficiency (McNaughton et al., 2007; Brock et al., 2019). No difference in volume comparison (discussed in Sect. 3.2) was found under those conditions, nor in previous missions with on average larger accumulation mode peaks (Fig. S4) hence we conclude that this is a valid assumption.

Another concern for airborne sampling with an AMS is the misalignment of the aerodynamic lens due to mechanical stress during flight. Such a misalignment will not necessarily be caught by the previously described calibrations, since they do not probe the full surface of the vaporizer, and since lens focusing can have some size-dependence. Hence for ATom 2-4, a particle beam width probe (Huffman et al., 2005) was flown and profiles of both the air and particle signal were taken at most airports during the mission, as shown in Fig. S9, directly confirming the lack of change in lens alignment.

During ATom, the AMS was operated in the fast mass spectrum mode (Kimmel et al., 2011), allowing for high-time-resolution measurements at 1 Hz. For every minute, AMS started with fast mass spectrum mode with the particle beam blocked (instrumental background measurement; 6 s) and then with the beam open (background plus ambient air and particles; 46 s) and ended with efficient particle time-of-flight (ePToF) mode (nominally 8 s), which measured speciated size distributions. The interpolated average of two consecutive background signals (beam closed) was subtracted from 1 s ambient signals (beam open). Also, fast blanks (20 s) were scheduled every 18 minutes by directing ambient air through a high-efficiency particulate air (HEPA) filter, serving to characterize the AMS zero (field background) and as a leak check downstream of the HEPA filter (Nault et al., 2018). It also serves as a frequent confirmation for the real-time continuous detection limits estimated using the method proposed in Drewnick et al. (2009). AMS data were reported at 1 s and 1 min time resolutions. For the 1



min product, the raw mass spectra were averaged prior to data reduction and analysis, which reduces
nonlinear spectral fitting noise for the least-squares error minimization method. This is observed because
a fit to the 1 min average spectrum has less fitting noise than the average of the fits to the 1 s spectra. In
the following analysis, the 1 min data product is used due to their improved signal-to-noise ratio (SNR).
Since the aerosol loadings were typically low and changed slowly in the global remote regions, longer
averaging times were used for some analyses. Continuous time-dependent detection limits (DLs) were
estimated using the method of Drewnick et al. (2009) and corrected by comparison with the periodic filter
blanks. The average DLs for the 1 min data were 76, 10, 6, 1, 7, 30 ng sm$^{-3}$ during ATom-1 and 133, 18,
9, 2, 10, 40 ng sm$^{-3}$ during ATom-2 for OA, SO$_4$, pNO$_3$, NH$_4$, Chl, and sea salt, respectively. Sea salt is
an important submicron aerosol component when sampling the marine boundary layer in ATom.
Although sea salt is not a standard AMS data product, in this study, we report AMS sea salt mass
concentrations with the method from Ovadnevaite et al., (2012) with a laboratory-calibrated response
factor, $9.8 \times 10^{-3}$, for the AMS sea salt marker Na$^{35}$Cl. Additional species were reported for ATom, with
DLs for MSA (methanesulfonic acid) and ChlO$_x$ (perchlorate) of 2 and 1 ng sm$^{-3}$ during ATom-1, 3 and
2 ng sm$^{-3}$ during ATom-2. Iodine and bromine were also quantified with DLs of 0.4 and 1.5 ng sm$^{-3}$
during ATom-1, 0.5 and 2 ng sm$^{-3}$ during ATom-2, as reported by Koenig et al. (2020). The variation in
AMS detection limits across species is mostly controlled by differences in background signals for
different ions. Many of these detection limits are lower than for typical AMS aircraft operation, especially
during the first several hours of each flight, due to the use of a cryopump in the CU AMS (Jayne, 2004;
Campuzano-Jost, 2012).

For the purpose of instrument comparisons, we estimate the aerosol volume based on the chemical

instruments ($V_{chem}$). The contribution of the AMS to $V_{chem}$ is determined from the AMS mass
concentrations by assuming volume additivity, with an average particle density estimated as (DeCarlo et
al., 2004; Salcedo et al., 2006):

$$\rho_m = \frac{OA + SO_4 + pNO_3 + NH_4 + Chl}{\frac{OA}{\rho_{OA}} + \frac{SO_4 + pNO_3 + NH_4}{1.75} + \frac{Chl}{1.52}} \tag{5}$$

The OA density ($\rho_{OA}$) is estimated with the AMS measured O/C and H/C atomic ratios of OA

using the parameterization of Kuwata et al. (2012) (when OA is under the detection limit and hence no



elemental ratios can be calculated, we assumed a default $\rho_{OA}$ of 1.7 g cm$^{-3}$ based on typical OA elemental
ratios found for concentrations close to the DL). The "improved-ambient" method was used for OA
elemental analysis (Canagaratna et al., 2015; Hu et al., 2018). The combined density of SO$_4$, NH$_4$, and
pNO$_3$ is assumed as 1.75 g cm$^{-3}$, an approximation from ammonium sulfate, ammonium bisulfate, and
ammonium nitrate (Sloane et al., 1991; Stein et al., 1994; Salcedo et al., 2006). The non-refractory
chloride density is assumed as 1.52 g cm$^{-3}$ based on ammonium chloride (Salcedo et al., 2006). The
frequency distributions of $\rho_m$ and $\rho_{OA}$ are summarized in Fig. S12. The mass-weighted average $\rho_m$ is
1.60 ± 0.14 g cm$^{-3}$ and 1.70 ± 0.10 g cm$^{-3}$, and $\rho_{OA}$ (averaged from above the concentrations above OA
DL) is 1.51 ± 0.19 g cm$^{-3}$ and 1.59 ± 0.24 g cm$^{-3}$ for ATom-1 and ATom-2, respectively. Negative AMS
mass concentrations exist at low concentrations since the AMS uses a difference measurement (signal
minus background). These negative AMS mass concentrations are kept as they are in deriving $V_{chem}$,
otherwise, a positive statistical bias would be introduced if a zero or a positive value was artificially
assigned to those data points.

### 2.5 Other aerosol measurements used in this study

The following instruments all sampled through the LARGE inlet, except Soluble Acidic Gases

and Aerosol (SAGA). The transmission efficiency for this inlet has been characterized as a function of
particle size by flying the NASA DC-8 in a previous campaign (McNaughton et al., 2007), demonstrating
a unity efficiency up to supermicron size ranges and reaching 50% at $d_{ta,air}$ of ~5 μm at the surface and
3.2 μm at 12 km. Hereafter, we refer to the 50% transmission diameter as $d_{50}$.

***Particle size spectrometers:*** Dry particle size distributions for $d_p$ from 2.7 nm to 4.8 μm were

reported at 1 Hz using three optical particle spectrometers, including a Nucleation-Mode Aerosol Size
Spectrometer (NMASS; custom-built; 0.003-0.06 μm) (Williamson et al., 2018), an Ultra-High
Sensitivity Aerosol Spectrometer (UHSAS; Droplet Measurement Technologies, Longmont, CO, USA;
0.06-1 μm) (Kupc et al., 2018), and a Laser Aerosol Spectrometer (LAS; LAS 3340, TSI, St. Paul, MN,
USA; 0.12-4.8 μm), all operated by NOAA Earth System Research Laboratory (ESRL). Two NMASS,
two UHSAS (during ATom-2 and -3, a 300 °C thermodenuder was installed upstream of the detector of
the second UHSAS to volatilize refractory components), and one LAS comprise the package of Aerosol





Microphysical Properties (AMP). Brock et al. (2019) discussed extensively the data inversion method to
merge the three non-thermally denuded size distributions into one. Hereafter, we refer to the non-
thermally denuded integrated volume (2.7 nm-4.8 µm) as the physical sizing-based volume ($V_{phys}$). AMP
performed well during ATom. Most relevant to the AMS size range, the UHSAS reported volume was
estimated to have an asymmetric uncertainty of +12.4%/-27.5% due to the differences in refractive index
($n$) between ambient particles and assumed ammonium sulfate particles ($n$ = 1.527, which is similar to
the refractive index found for aged ambient OA (Aldhaif et al., 2018)). This uncertainty range is estimated
to be between 1σ and 2σ depending on the conditions. Here we assume that it represents 1.5σ when using
it for uncertainty analyses.

***SP2:*** Refractory Black Carbon (rBC, as defined in Petzold et al. (2013)) mass concentrations in

the accumulation mode size range were measured by the NOAA Single Particle Soot Photometer (SP2;
Droplet Measurement Technologies, Longmont, CO, USA) (Schwarz et al., 2010b; Katich et al., 2018).
The ATom SP2 detection system was operated as in Schwarz et al. (2010a) with a size range for rBC
mass of $d_{ve}$ ~90-550 nm (Schwarz et al., 2010b). This size range typically contains ~90% of the total rBC
mass in the ambient accumulation mode (Schwarz et al., 2008; Shiraiwa et al., 2008).

***PALMS:*** The Particle Analysis by Laser Mass Spectrometry (PALMS) is a single-particle laser-

ablation/ionization mass spectrometer instrument that measures size-resolved ($d_p$ ~ 0.1-5 µm) particle
chemical composition with fast response (Thomson et al., 2000; Murphy et al., 2006). Particle mass
concentrations can be derived as a function of size when mapping the PALMS chemical composition to
the size distributions reported from the UHSAS and LAS, which is referred to as the PALMS-AMP
products (Froyd et al., 2019). In this study, we focus on the different particle size ranges observed by
PALMS and AMS, to illustrate the strengths and applications of the two aerosol composition instruments
onboard the DC-8.

PALMS is the most complex of the chemical composition instruments used in ATom. It is a single-

particle based instrument with both a very steep detection efficiency vs. particle size in the smaller particle
range and the ability to measure much larger particles than the AMS. While the total reported mass (with
some density uncertainty) of the PALMS-AMP products will always match the physical volume
measurement over the range that PALMS reports (100-5000 nm $d_p$), the uneven sampling data coverage



of particles across each size bin, as well as the broadness of the bins chosen for PALMS-AMP analysis,
can lead to a chemical bias if composition gradients exist within a bin (Fig. S13). Therefore, care must be
taken to balance statistical representativeness against the need for unvarying particle composition across
the size range over which those statistics are obtained (Froyd et al., 2019).

For intercomparisons we characterize the specific size range over which the PALMS can obtain

sufficient chemical information over a given time period under the ATom conditions, which is mainly
limited by particle statistics. If zero or a very low number of particles is sampled for a given AMP size
bin and time period, there is no real information being captured for characterizing the composition of the
particles in that bin. That is true even if the AMP volume in that bin is assigned a composition by
extrapolating the composition of larger or smaller particles. Therefore, we derived the PALMS detected
particle numbers based on the raw AMP size resolution (20 bins/decade, 34 bins in total above 100 nm $d_p$
for the size range that PALMS-AMP reports) to avoid the assumption of homogeneous chemical
composition within four broader bins in Froyd et al. (2019). This provides an alternative illustration of
PALMS size coverage and introduces a method that is applicable to other single-particle mass
spectrometers or other particle-counting based chemical instruments. A sensitivity test was carried out at
various size resolutions, shown as Fig. S14, with more detected particles per bin for larger bins, as
expected. Importantly, the 20 bins/decade size resolution of AMP is preferable as it makes the results
directly comparable to the other aerosol instruments. The probability of detecting on average one valid
particle per AMP size bin in the PALMS is very low below ~160 nm and above 1000 nm over a typical
3 min analysis period (Fig. S14). As altitude increases in the free troposphere, the size distribution often
shifts to smaller diameters (Williamson et al., 2019), thus we expand the 1D profile in Fig. S14 to include
the altitude dependence (Fig. S15). The results are shown at 3 min, 60 min intervals, as well as campaign-
wide, since the 3 min timescale is most relevant for high time resolution airborne analyses while the
longer ones are relevant to averages by altitude in a latitude band and similar analyses that group data
together from different time periods.

Based on Froyd et al. (2019), we assume that if PALMS detects N = 1 particle in a given AMP

size bin, the composition of the bin is fully characterized. This particle number corresponds to N = 5
particles for 4 bins/decade, which is a reasonable number for the particle composition to be reasonably





represented by the particles captured, with a resolution over which composition changes may happen in
the real atmosphere (Zhang et al., 2004). For simplicity, we scale the fraction of the particles contributing
information content linearly for conditions with N < 1.
*SAGA:* gas-phase HNO$_3$ plus particulate inorganic nitrate, and sulfate were measured online with
the University of New Hampshire (UNH) SAGA mist chamber (MC) ion chromatography (IC) at a time
resolution of ~80 s. Water-soluble chemical species were also measured offline by collecting particles
with Zefluor filters (9 cm diameter, 1 mm thick, and 1 μm pore size, from MilliporeSigma Corp.,
Burlington, MA, USA) with subsequent procedures as described by Dibb et al. (1999, 2000) and Heim et
al. (2020). In brief, filter samples were collected during level portions of each flight, stored over dry ice,
extracted with ultrapure water, and sent back to the lab in UNH for IC analysis to quantify more species
than the MC (Dibb, 2019).
SAGA filters were sampled from the UNH inlet with an estimated cutoff size of  4.1 μm ($d_{ta,sea,50}$)
at the surface and 2.6 μm ($d_{ta,air,50}$) at 12 km (McNaughton et al., 2007). The SAGA MC sampled from a
glass-coated (vapor deposited) manifold (8 cm inner diameter) with high airflow (on the order of 2000 sL
m$^{-3}$ at low altitude) (as shown in Fig. S16). The diffuser type configuration at the manifold entrance boosts
airflow and the surrounding piece at the pipe tip excludes cloud droplets and giant sea salt particles (Talbot
et al., 2003). The in-cabin part of the pipe till MC was heated to 50 °C to minimize HNO$_3$ wall deposition,
although sampled air $T$ is assumed to be the same as ambient due to the high airflow and short residence
time (~0.2 s). A small glass tube from MC, which is sealed at the bottom and opens a small hole on the
downstream side, sticks down into the manifold. This configuration provides a particle cutoff size of
~1μm ($d_{ta,sea,50}$) at the surface and lower at higher altitudes (van Donkelaar et al., 2008).
To be compared with other ATom aerosol measurements, the pressure-dependent SAGA MC and
filter inlet transmissions are calculated based on the ATom conditions and summarized in the SI as Fig.
S17 and S18, respectively.
***Use of data from other instruments for $V_{chem}$:*** $V_{phys}$ includes refractory species, such as rBC, sea
salt, and dust, and thus their volumes need to be added to the AMS non-refractory volume before
comparison. rBC volume is estimated from SP2 mass measurements (Katich et al., 2018) with a density
of 1.77 g cm$^{-3}$ (Park et al., 2004). The sea salt volume is estimated from its AMS mass concentration with



a density of 1.45 g cm$^{-3}$, assuming particles had not fully effloresced prior to detection (Froyd et al.,
2019). Sea salt is typically externally mixed with sulfate-organic-nitrate particles (Froyd et al., 2019),
therefore, it is not routinely considered in the AMS aerosol density estimation (i.e., in Eq. 5). The
exclusion of dust in the volume closure is reasonable in general based on the results in Sect. 3.2 due to
the limited impacts from dust for ATom, on average 1.1 ± 4.3 % (median = 0.0 %) of the AMS observed
volume, but it can contribute as high as 95% for occasional short plumes encountered in ATom-2 (Fig.
S19) (Froyd et al., 2019). Besides, we exclude the last ATom-1 research flight (a transit flight in the
continental U.S. from Minneapolis, MN to Palmdale, CA, different from the remote marine atmosphere
of the other ATom flights) and <10 min of sampling impacted by volcanic ash near Hawaii in ATom-2
(Research Flight 203, Jan 30, 2017). As discussed above, we use 1-min AMS data for intercomparison,
and 1 s $V_{phys}$ is averaged to the same time scale. There may be a minor bias introduced from this approach
since AMS periodic blank measurements exclude some 1-sec data points from the AMS but not from
$V_{phys}$ (~3% of the total 1-sec $V_{phys}$ points), and similarly, some data are removed from the sizing
measurements due to cloud masking but not for the AMS (13%, discussed below in Sect. 3.2). In this
study, the particle volume is reported in units of µm$^3$ scm$^{-3}$, where scm$^{-3}$ are cubic centimeters of air under
STP.

**2.6 Summary of the ATom aerosol size distribution and instrument size ranges**

Fig. 1 summarizes the ATom-2 campaign averaged number and volume size distributions from

AMP and compares it to the subranges observed from several ATom aerosol instruments, to provide
context for this study and future instrument comparisons based on the ATom dataset. The upper cut sizes
for LAS, SAGA MC, and filter, determined from their inlets, move towards smaller particles at higher
altitudes, thus the size ranges plotted in Fig. 1 for these instruments are the best-case scenario (in the
planetary boundary layer). In contrast, the AMS transmission stays the same up to ~9 km. Based on Fig.
1, the AMS size range is more closely comparable to SAGA MC, and comparison to all the other
instruments requires considering the different size ranges. Therefore, accurately characterizing AMS
transmission is a prerequisite for quantitative instrumental intercomparisons. While the focus of this work
is on instrument comparisons, we want to emphasize that a properly characterized size cut is also





important for model comparisons and that the size bins used in most global models, typically reported as
$d_p$, vary widely (Hodzic et al., 2020).

## 3 Results and Discussion

### 3.1 AMS transmission

AMP gives nearly unity detection efficiency of the particles (not lost in the inlet) from ~5 nm to
~4 µm ($d_p$) at sea level, and 50% transmission at 2.7 nm and 4.8 µm (inlet-limited), of which AMS, SAGA
MC, PALMS, and SP2 observe a subrange, as shown in Fig. 1 (McNaughton et al., 2007; Brock et al.,
2019). Therefore, the volume derived from the AMP size distributions ($V_{phys}$) can be used as the basis for
intercomparisons. Characterizing AMS transmission ($E_L$) is critical for a meaningful comparison of $V_{phys}$
vs. $V_{chem}$.
AMS transmission (always specified vs. $d_{va}$) can be quite variable between instruments, and can
also change for a specific AMS in time, so it is critical to characterize the transmission in the field for
meaningful instrumental intercomparisons (Liu et al., 2007; Knote et al., 2011; Hu et al., 2017; Nault et
al., 2018). During ATom, the large particle region (~500-1200 nm, $d_{va}$) of the CU AMS transmission was
calibrated in the field (Fig. 2), as discussed in Sect. 2.4. A fit to the multi-size field calibrations indicates
a 100% transmission at $d_{va}$ of ~483 nm (1σ range: 445-525 nm) and a 0% transmission at ~1175 nm
(1112-1241 nm), with 50% transmission at 754 nm. This transmission was stable throughout the ATom-
1 and -2 deployments. Other than new particle formation and growth events, the small particle end of the
transmission curve is less critical in determining submicron aerosol volume since volume is normally
dominated by the accumulation mode (which normally refers to the range 100-1000 nm $d_{ta}$) (Seinfeld and
Pandis, 2016) instead of the Aitken mode (10-100 nm $d_{ta}$). Brock et al. (2019) found the accumulation
mode during ATom to be 60-500 nm $d_p$, equivalent to 93-674 nm in $d_{ta,sea}$, as remote particles were far
away from sources of precursor gases that could sustain growth to larger sizes. Results from previous
measurements (Zhang et al., 2004; Knote et al., 2011) were used to estimate the small particle
transmission, with 0% at 35 nm and 100% at 100 nm. Sensitivity tests on the small particle transmission
points (Sect. 3.3 below) confirm a lack of impact on the volume comparison for ATom conditions. AMS



transmission curves for all ATom campaigns are shown in Fig. 3. Importantly, AMS transmission
improved noticeably for ATom-4 compared to the prior ATom legs, possibly due to small changes in the
inlet during reassembly. This shows the importance of characterizing $E_L$ for each campaign for
quantitative intercomparisons. Similar changes have been observed in the past for other aircraft and
ground campaigns.

**3.2 Comparison of AMS vs. standard PM$_1$ size cuts**

AMS is often described as an approximate "PM$_1$" or "submicron" instrument. Since the standard

definition of PM$_1$ is based on devices that impose an aerodynamic diameter ($d_{ta}$) cut under ground-level
pressure, temperature (e.g., defined at $T$ = 293.15 K and $P$ = 1013 mbar (Marple et al., 1991)), and
humidity, the equivalent AMS transmission in $d_{ta}$ depends on particle density and composition, as well
as the $E_L$ of the specific AMS for a given study. The careful transmission calibrations and extensive
sampling of ATom allow more precise characterization of this cutoff size for the CU aircraft AMS and
remote aerosols.

For aircraft sampling where a submicron cut is desired (not including the AMS), the single 1 μm

stage from a micro-orifice uniform deposit impactor (MOUDI) (Marple et al., 1991, 2014) is often used
(e.g., (Peltier et al., 2008; Brock et al., 2011; Guo et al., 2016)) to preselect submicron particles
(transmission shown in Fig. S20). Here, we choose MOUDI instead of SAGA MC, also known as a
submicron cut instrument deployed for aircraft studies, due to the lack of a published transmission curve
for SAGA MC. Due to the higher temperature in cabin vs. ambient air (Guo et al., 2016), the MOUDI
impactor (operating at cabin $T$ and ambient $P$) is expected to size-select dry particles, similar to the AMS.
The impactor provides a nominal PM$_1$ cut at $T$ = 293.15 K and $P$ = 1013 mbar but the $d_{ta,50}$ for a given
particle is pressure- and temperature-dependent, and thus varies with altitude. For instance, at an aerosol
density of 1.7 g cm$^{-3}$ (the ATom-2 campaign average), $d_{ta,air,50}$ drops from 1 μm to 912 nm at 6 km, and
to 686 nm at 12 km height, based on the U.S. standard atmosphere (NOAA, NASA, U. S. Air Force,
1976), as shown in Fig. 3. Even lower cut sizes, 752 nm at 6 km and 400 nm at 12 km, are expected if the
impactor was operated under ambient $T$ (not typically done, and best avoided for an optimal particle cut;
summarized in Table S1). Hence, the deviation from the nominal 1 μm cut size can be very significant at





high altitude (although it could in principle be modulated by changing the flow rate vs. altitude). The
pressure-dependent diffusion loss of small particles for MOUDI is estimated using the inlet system
onboard NCAR/NSF C-130 from Guo et al. (2016), a ~2.5 m tubing with an inner diameter of ~1.1 cm.
Given a flow rate of 30 L m$^{-3}$, Reynolds number is 3858 at sea level and increases with altitude, indicating
a turbulent flow in the inlet.

If we compare the AMS transmission to ground-level based dry $d_{ta}$ (using a dry particle density

of 1.7 g cm$^{-3}$ to calculate $d_{ta}$ from $d_{va}$), the ATom-2 / 3 / 4 $d_{ta,sea,50}$ are 599 nm, 615 nm, and 758 nm,
respectively (the $d_{ta,air,50}$ are higher and listed in Table S1; for example, $d_{ta,air,50}$ is 782 nm and 837 nm at
6 km and 12km, respectively for ATom-4). Thus the cutoff size of the AMS in ATom is more stringent
than a MOUDI nominal PM$_1$ cut at the surface and 6 km, and less stringent at the higher altitudes in
ATom-4. Importantly, the AMS transmission stays constant up to ~ 9 km in altitude for the implemented
PCI. No in-field characterization of the AMS transmission at higher altitudes (when inlet pressure slips)
was performed, but laboratory calibration shows no change in transmission at 710 nm $d_{va}$ at the max
altitude inlet pressure (1.05 Torr).

For ground studies, URG PM$_1$ standard cut (model: URG-2000-30EHB) and sharp cut (model:

SCC 2.229) cyclones are widely used for non-AMS instruments. The estimated diffusion loss of small
particles in the URG cyclones was negligible (e.g., 5% loss at $d_{ta,sea}$ = 5 nm and less loss expected at
larger sizes), calculated with a nominal flow rate of 16.7 L m$^{-3}$ and assumed cyclone internal dimensions,
0.50 inch (1.27 cm) in diameter and 50 cm in length (Reynolds number = 2100, indicating a likely
turbulent flow). The two cyclones offer cutoff sizes at 1 μm at $T$ = 293.15 K and $P$ = 1013 mbar (Fig.
S20), and smaller cuts when such cyclones are deployed at lower ambient pressure and the nominal
volumetric flow, e.g., at a mountain site.

One additional complexity arises since the standard PM$_1$ cut made with URG cyclones  are under

ambient humidity conditions (i.e., particles are not dried prior to sampling). Thus, the equivalent dry
particle cut size is below 1μm at sea level and depends on the amount of liquid water associated with the
particles. For the ATom conditions, particle size shrinks on average ~20% (assuming a complete loss of
the predicted particle liquid water content from the higher ambient RH, mean/median(±SD) =
40/36(±29)% to the lower inlet RH, 10/0.4/(±21)%; Fig. S1c-d) and the frequency distribution plots are



shown as Fig. S21 (SD stands for standard deviation). While AMS transmission is characterized with dry
particles, a smaller difference between the AMS transmissions and the cyclone transmissions is expected,
compared to Fig. 3. Taking the estimated ~20% shrinkage in particle size from drying in the sample line
(for the ATom-1 and -2 conditions), the AMS transmission would be equivalent to a standard $PM_{0.75}$ and
a $PM_{0.95}$ cut during ATom-2 and -4 respectively in terms of ambient aerosol size.

Since aerosol density affects the conversion between $d_{va}$ and $d_{ta}$ (Eqs. 1-2), a higher AMS $d_{ta,50}$ is

expected if sampling aerosols with lower densities than the ATom-2 campaign average of 1.70 g cm$^{-3}$. To
illustrate this point further, results based on an assumed 0.9 g cm$^{-3}$ aerosol density, typical of hydrocarbon-
like OA from lubricating oil or oleic acid as cooking aerosol surrogate (Kuwata et al., 2012; Herring et
al., 2015), are shown in SI as Fig. S22b. In this case, the ATom-2 and ATom-4 AMS $d_{ta,sea,50}$ increase to
789 nm and 1006 nm, respectively, making the ATom-4 AMS a dry $PM_1$ cut when performing
experiments with those aerosols.

It is also useful to compare the sharpness of the different transmission curves. The sharpness of

transmission is commonly defined as $(d_{ta,16}/d_{ta,84})^{0.5}$, where $d_{ta,16}$ and $d_{ta,84}$ are particle aerodynamic
diameters at 84% and 16% transmissions (Peters et al., 2001). The sharpness of the AMS transmission
profiles are similar to that of a URG $PM_1$ standard cut cyclone; 1.34 in ATom-2 and 1.49 in ATom-4
compared to 1.35 and 1.17 of the URG standard cut and sharp cut cyclones (a lower number indicates a
sharper cut). The MOUDI 1μm stage impactor provides the sharpest cut at 1.12 at sea level but the
sharpness decreases at higher altitudes, 1.15 at 6 km and 1.22 at 12 km.

Including all effects, the CU aircraft AMS was approximately equivalent to a standard ground-

level $PM_{0.75}$ instrument during ATom-2 and a $PM_{0.95}$ instrument during ATom-4. For laboratory or field
experiments with oily particles with an aerosol density of 0.9 g cm$^{-3}$, the same AMS would be a $PM_{0.79}$
or $PM_{1.0}$ instrument in terms of dry aerosol size.

### 3.3 Volume closure

AMS observes a fraction of $V_{phys}$ (Figs. 1 & 3). AMS transmission vs. $d_p$ was calculated based on

the calibrated transmission vs. $d_{va}$ (Eq. 1) and the AMS estimated time-dependent $\rho_m$ (Eq. 5), and applied
to the AMP size distributions used to derive $V_{phys}$. This volume is referred to as $V_{phys,TC}$ (the AMS-





transmission-corrected $V_{phys}$). The comparisons between $V_{phys,TC}$ and $V_{chem}$ for ATom-1 and -2 are shown
in Fig. 4. Good agreement is observed, with the data points distributed around the 1:1 line over a three
order-of-magnitude range of concentrations. For ATom-1 the regression slope is 0.96 and $r^2$ is 0.95. The
larger volume concentrations were generally detected in the boundary layer. Time averaging reduces
random noise (more dominant at smaller volumes), as evidenced when comparing this analysis for 1, 5,
and 10 min averages (Fig. S23). The slightly worse fitting slope of 1.09 and $r^2$ of 0.93 in ATom-2 may be
due to the larger contribution of sea salt in ATom-2 in the boundary layer (Hodzic et al., 2020) and hence
the larger uncertainty in applying the AMS size cut. To illustrate the impacts of sea salt, we replotted the
comparisons (Fig. 4a-b) colored by sea salt shown as Fig. S24a-b, which suggests that some outliers in
ATom-2 are observed at high sea salt concentrations. We also investigate the potential differences in the
data products due to the differences in raw data processing criteria for cloud artifacts between AMS and
NOAA size spectrometers and find no clear evidence (Fig. S24c-d). Furthermore, we confirm that
excluding submicron dust volume is reasonable; only a few outliers have noticeably higher contributions
from dust (Fig. S24e-f).
Species density is used to convert the AMS mass to volume concentrations and thus affects the
volume comparison. As discussed above (Fig. S12), $\rho_{OA}$ in this study is estimated with the
parameterization method of Kuwata et al. (2012). The $\rho_{OA}$ parameterization method from Kuwata et al.,
(2012) was validated up to 1.9 g cm$^{-3}$ (i.e., oxalic acid) and the lab generated SOA in that study had up to
1.46 g cm$^{-3}$ $\rho_{OA}$ with an O/C of 0.72. The estimated ATom-1 and -2 $\rho_{OA}$ is close to that of succinic acid,
1.57 g cm$^{-3}$, that has a similar O/C ratio (ATom-1 and -2 vs. succinic acid: 1.05 ± 0.44 vs. 1.0), and falls
into the observed $\rho_{OA}$ density range, 1.5-1.7 g cm$^{-3}$, for low mass concentrations of SOA (< 3 µg m$^{-3}$, as
the most cases in ATom), made from α-pinene and ozone from a chamber study (Shilling et al., 2009).
However, $\rho_{OA}$ estimated from PALMS, 1.35-1.45 g cm$^{-3}$ (Froyd et al., 2019), is ~0.2 g cm$^{-3}$ lower than
that estimated from AMS, for reasons that are not yet understood. As a sensitivity check, we recalculate
$V_{chem}$ by subtracting 0.2 g cm$^{-3}$ from the AMS estimated $\rho_{OA}$ (Fig. S25). Compared to the base cases (Fig.
4a-b), the $r^2$ values barely change and the slopes increase by 5% or 8% due to the higher estimated OA
volume in $V_{chem}$. Therefore, this uncertainty is below 10% and does not undermine the agreement within
the uncertainties between $V_{chem}$ and $V_{phys,TC}$.



To illustrate that applying the AMS transmission to $V_{phys}$ is a prerequisite for a meaningful

comparison, Fig. 4c illustrates the volume closure for a research flight in ATom-2 (RF208, Feb 15 2017,
from Ascension to the Azores), in which the contribution of supermicron particles to total volume is
significant. Although $V_{phys}$ was, in general, several times larger than $V_{chem}$ when the DC-8 flew at lower
altitudes (below ~3 km), $V_{phys,TC}$ agrees very well with $V_{chem}$, with a regression slope of 1.04 and an $r^2$ of
0.97. To examine if applying the AMS transmission introduces a systematic bias, Fig. 4a-b was replotted,
colored by the removed fraction of $V_{phys}$, in SI as Fig. S26. The binned data points at 20% intervals show
little difference, suggesting that no significant bias is arising for this reason for both ATom-1 and -2. An
exception is the 80-100% bin for ATom-2 due to some outliers with high sea salt as shown in Fig. S26b
and possibly the increased statistical noise, with only 25% of the data points in this bin compared to
ATom-1.

Box plots, regressions, and correlations were carried out for the separate datasets in each bin of

removed $V_{phys}$, as shown in Fig. 5a-c. For the combined ATom-1 and -2 data (Fig. 5a), the majority of the
volume ratios are distributed around the 1:1 line and within the combined systematic uncertainty range
(combined 2σ of AMS and UHSAS, the size spectrometer that overlaps most with the AMS, see Fig. 1).
If using the UHSAS data product alone and applying the AMS transmission, the resulting volume is on
average 93 ± 9 % in ATom-1 and 87 ± 14 % in ATom-2 compared to $V_{phys,TC}$. Therefore, the UHSAS
uncertainty is representative of that of $V_{phys,TC}$. The $V_{phys}$ uncertainty depends on particle size range or
mode (see Table 1 in Brock et al., (2019)) and the random uncertainty in $V_{phys}$ is expected to be smoothed
out with longer averaging time scales. All five bins show high correlations with $r^2$ of 0.79-0.96, with a
lower correlation at the 80-100% $V_{phys}$ removal bin. The smallest slope of 0.84 is also seen at this bin,
where the largest discrepancy is expected due to the combined sharpness of the decreasing AMS
transmission for larger particles and the rising tail of coarse mode particles into the submicron size range
(e.g., the AMS transmission excludes on average 89% of the total sea salt volume sampled during ATom-
2). When investigating ATom-1 and ATom-2 independently, ATom-1 averages are slightly below unity
but consistent throughout the five bins (Fig. 5b), and ATom-2 shows an increasing bias above 60% $V_{phys}$
removal (again likely due to the much higher sea salt fractional contribution for this campaign). Only the
80-100% bin in ATom-2 has substantial data outside the 2σ uncertainty range. Overall, the above results



suggest the in-field characterized AMS transmission is robust for the various conditions encountered in
the ATom-1 and -2 studies.

While binning the data is useful for exploring possible systematic biases, looking at the overall

deviations of the individual measurements allows us to explore to what extent the reported instrument
uncertainties are consistent with the ATom dataset. Fig. 5d-i shows the frequency distributions of the
volume ratio, $V_{chem}$ / $V_{phys,TC}$, together with the combined 2σ accuracy of AMS and UHSAS. The ATom-
2 data distribution is slightly broader than ATom-1 partly owing to the larger precision error (e.g., when
mass concentration is within three times of DLs) associated with the lower submicron mass
concentrations, 0.38 vs. 0.50 µg m$^{-3}$. Longer averaging time can deemphasize the precision errors,
especially for a dataset like ATom with few sharp plumes. Thus we plot the volume ratio at three time
scales, 1 min, 5 min, and 10 min. It shows a clear improvement in the spread of the ratio as the averaging
time scale increases, with the 10 min data being consistent with the reported accuracies. This supports
the good quality and consistency of the ATom aerosol dataset, and it also supports the reported AMS
accuracies.

## 3.4 Sensitivity tests to AMS transmission

The above discussion demonstrates the critical role of well-characterized AMS transmission for

meaningful volume intercomparison. In this section, we aim to quantify the impact of the AMS
transmission on the volume comparison by artificially adjusting the transmission with a series of
sensitivity tests. As shown in Fig. 6a, the AMS transmission can be characterized by four "anchoring"
particle sizes, representing 0% and 100% transmissions at both ends. During ATom-1 and -2, these
anchoring sizes (in $d_{va}$) were estimated as (i) 35 nm, (ii) 100 nm, (iii) 482 nm, and (iv) 1175 nm,
respectively, as discussed above (Fig. 2). Uncertainty ranges are estimated for the latter two sizes from
the ATom calibrations and shown in Fig. 6d-e. We alter one anchoring size at a time, recalculate $V_{phys,TC}$,
and re-compare to $V_{chem}$, which is kept unchanged. The resulting slopes and r$^2$ are summarized in Fig. 6.
The adjustments at the two lower anchoring sizes, up to ± 25 nm at 35 nm and ± 50 nm at 100 nm, have
a negligible impact on the volume comparison due to the small volume/mass concentrations at these sizes
during ATom (e.g., Fig. 3), except for the unrealistic 50 nm decrease at 100 nm (the second anchoring

point). In contrast, a dependency of the fitting results on the details of the AMS transmission curve for large particles is observed. For the third anchoring point, corresponding to the largest particles with 100% transmission (Fig. 6d), a smaller $d_{va}$ excludes more $V_{phys}$ and results in a higher slope. For example, at the lower one SD limit $d_{va}$ of 445 nm, the fitting slopes increase from 0.97 to 1.01 for ATom-1 and 1.09 to 1.12 for ATom-2. These small changes in slope are the largest among the four anchoring points, and they are statistically significant because the changes are one magnitude higher than the fitting 1σ uncertainties of the slopes (~0.03 vs. ~0.004). In all the cases investigated, $r^2$ barely changes.

It is also of interest to compare the results if we had assumed that AMS literature transmission curves applied to this study. Here we test the commonly used transmission curves of Liu et al., (2007) and Hu et al., (2017). The four anchoring sizes, all in nm, in Liu et al., (2007) (Hu et al., (2017) in parentheses) are (i) 50 (40), (ii) 150 (100), (iii) 300 (500), and (iv) 1400 (1500, estimated by fitting). The regression slopes with the Liu curve (the Hu curve) are 1.18 and 1.23 (0.94 and 0.96) in ATom-1 and -2, respectively, compared to 0.96 and 1.09 derived from applying the ATom-1 and -2 transmission (Fig. 4). In summary, the above results suggest: (1) The volume closure is relatively insensitive to the uncertainties of the AMS transmission curve characterized in this study; (2) Use of transmission curves from the literature for uncharacterized instruments can result in substantial deviations (which may then be incorrectly attributed to changes in *CE* or *RIE*); (3) The large particle region of the AMS transmission curve is more important than the small particle region for ATom-1 and -2; and (4) The point (iii) with 100% transmission size for large particles (482 nm in this case) is the most important calibration, due to the dominance of the accumulation mode mass for the submicron size range.

### 3.5 Characterization of the AMS observable particle fraction during ATom vs. the standard ground-based and aircraft-based $PM_1$ definition

It is of interest to compare the fraction of the volume detected by the AMS for ATom vs. what a standard ground-level $PM_1$ (the most common definition of "submicron") instrument would detect. In this study, we use the standard cut URG cyclone operating at the surface ambient humidity as the reference, simulating its operation at ground sites at different altitudes (e.g., sea level and mountain sites). As discussed above, both the AMS and the AMP size distributions measure dry particles while the "standard" $PM_1$ is defined with practical size-selection under ambient humidity. To account for the difference, the





URG transmission is applied to the estimated ambient particle size before losing liquid water content (the
effect of water on $\rho_p$ is also considered) (DeCarlo et al., 2004). We assume no size dependence for $\rho_p$ or
the volume fraction of liquid water content for the submicron aerosols. Ambient $P$ and $T$ from ATom are
applied to the URG transmission to account for the shift at non-STP conditions, which is relevant when
operating such a cyclone at higher altitudes e.g., a mountain site. The results of applying the AMS and
URG $PM_1$ standard cut cyclone transmissions to $V_{phys}$ are shown in  Fig. 7. AMS observed on average 96
± 16% (median 96%) and 94 ± 12% (median 94%) of the volumes that would transmit through a ground-
level URG $PM_1$ cyclone in ATom-1 and -2, respectively. Although we previously concluded that the
AMS was approximately an equivalent ground-level $PM_{0.75}$ instrument in ATom-1 and -2, the difference
in collected volume is only ~5%. This is because the submicron volume size distribution peaked around
300 nm ($d_{ta}$; see Fig. 3 for example), where AMS transmission is ~100%, and also due to the effect of
liquid water on particle size.

Next, we compared the submicron volumes observed from the CU AMS and a MOUDI 1 µm

stage impactor during aircraft studies, using the ATom conditions (Fig. 7c & 7d). The two inlets size-
select dry particles due to sample line heating. AMS observed 87% and 83% by means, 90% and 85% by
medians, in ATom-1 and -2 of that from an airborne MOUDI impactor, lower than the ratios when
comparing to the URG $PM_1$ cyclones for two reasons: the smaller cutoff size of URG vs. MOUDI due to
particle water and lower operating $T$ for URG (which relates to air viscosity). We also compared the
$V_{phys,TC}$ to the (total) $V_{phys}$ (Fig. 7a & 7b). AMS collected 68% by means (the same for ATom-1 and -2,
and 78% in ATom-1 and 71% in ATom-2 by medians) of $V_{phys}$; in other words, 32% of $V_{phys}$ was excluded
by applying the AMS transmission. For both ATom-1 and -2, there was considerable variability on the
fraction of $V_{phys}$ removed to obtain $V_{phys,TC}$, which spanned the range from 0% to 100% removal, thus
providing a good scenario of testing the AMS transmission. Nevertheless, this data shows that on average
the AMS captured the submicron range well, as shown in Fig. 4, and that the comparisons presented here
are meaningful for a wide range of scenarios.



## 3.6 Characterization of the observable particle populations for different chemical instruments

The different parts of the aerosol population included in different measurements and models make comparisons of aerosol species inherently more complex than for gas-phase species. In this section, we characterize the size ranges that contribute information to each composition measurement. Importantly, only the particle ranges are illustrated, irrespective of the properties of each chemical detector (e.g., species measured, detection limits, etc.). Speciated particle mass concentrations can be derived by sampling the bulk aerosol using a size cut. For example, MOUDI 1 µm stage impactor and SAGA MC are suitable for size-selecting submicron range (Fig. S17). With a wider coverage expanding to supermicron sizes, SAGA filters measure up to $d_{ta,sea}$ of 4.1 µm and their estimated altitude-dependent transmissions for the ATom conditions are shown in Fig. S18. Speciated mass concentrations can also be derived as a function of size by mapping the PALMS single-particle chemical composition onto an independent physical size distribution measurement (in case of ATom the AMP size distribution products described in Brock et al. (2019)) (Froyd et al., 2019), and PALMS-AMP derived sulfate and organic mass concentrations have recently been reported to the NASA ATom archive (Wofsy et al., 2018).

Fig. 8 summarizes the approximate fractions of the volume and number distributions that each ATom instrument observed for ATom-2 (Fig. S28 shows ATom-1). A MOUDI 1 µm stage impactor is also included for comparison. SAGA filters collect nearly the entire total volume. The vertical profiles of volume size distributions collected by AMS and MOUDI are similar and converge at higher altitudes due to the shift in the MOUDI cutoff size. Both AMS and PALMS capture the accumulation mode, which often dominates particle mass, and thus agreement of the reported submicron concentrations should be expected under such conditions. The AMS samples contain chemical information about smaller particles that are typically absent from the PALMS data (Williamson et al., 2019). Conversely, the PALMS samples a significant fraction of the supermicron mode beyond the transmission range of the AMS. The PALMS-AMP at the reported AMP size resolution and 3 min time resolution is shown in Fig. 8 (and Fig S28), and similar plots for other size and time resolutions are shown in Fig. S29 and S30. 3 min corresponds to ~36 km horizontal distances and ~1.5 km vertical distances during ATom profiles and thus is a reasonable basis for comparison.



It is also of interest to quantify what fraction of the particle number is represented by each

instrument's data. For instance, the composition relevant to calculations of cloud condensation nuclei
(CCN) number concentrations would be dominated by small particles. The number fractions have
somewhat different meanings for the instruments. PALMS, when merged with size distribution
measurements, can quantify the number of particles of various types as a function of size. For the other
(bulk) instruments, the number fraction merely represents the number of particles in the size range where
mass is measured. Unlike the volume case, where the size distribution is dominated by the accumulation
and coarse modes, the number size distribution in ATom was dominated by the nucleation and Aitken
mode particles. In ATom-1 and -2, the SAGA filters, MOUDI, AMS, and PALMS-AMP (based on AMP
size resolution and 3 min time resolution) characterize the chemical composition on average of 96%
(median 99.9%), 78% (87%), 68% (74%), and 54% (55%) of $V_{phys}$ (total AMP particle volume), and 98%
(99%), 89% (93%), 41% (41%), and 5% (1%) of the total AMP particle number, respectively. The size
range above $d_p = 100$ nm, for which PALMS-AMP (Froyd et al., 2019) reports chemical products
(partially by extrapolating composition measurements of others sizes, especially at higher time resolutions
and lower concentrations), covers 76% (83%) and 11% (5%) of the AMP volume and number,
respectively.

To complete the illustration of the coverage of the previously discussed instruments, the vertical

profiles of observed volume fractions, in both the submicron range and the full AMP size range, are
summarized in Fig. 9 (and the statistics summarized in Table S2 in SI). For the submicron measurements,
AMS is highly comparable to the URG PM$_1$ standard cut cyclone, MOUDI 1 µm stage impactor, and
SAGA MC. More particle volume is observed by AMS as altitude increases, due to the relatively constant
AMS lens transmission (that always operates in the free molecular regime) and the smaller aerodynamic
cutoff sizes for the other three inlets (that operate at ambient $P$). For the AMP size range, similar
increasing fractions of $V_{phys}$ as a function of altitude are observed in all the panels, except for PALMS-
AMP, due to the larger fraction of the aerosol population at smaller diameters aloft than at the surface
(Fig. 8) (Williamson et al. 2019). PALMS excels in the lower 2 km of the atmosphere where it
characterizes most of the volume, while the submicron instruments only capture ~40-50 %. This clearly
shows the heterogeneity and complementarity between PALMS-AMP and the other submicron bulk





measurements as a function of altitude. The differences between the 3 min characterization and the PALMS-AMP products are greatly reduced by averaging to 60 min.

In summary, outside dust or biomass burning plumes, the particle volume sampled by AMS is within $97 \pm 14\%$ compared to SAGA MC, for which the difference disappears for the higher altitude legs, and $85 \pm 10\%$ of an airborne dry $PM_1$ measurement, a MOUDI impactor often used in aircraft. AMS and PALMS particle compositional data overlap for a large part of the volume distribution in ATom, and they complement each other at the ends of the distribution (the statistics of the overlap are listed in Table S2). Last but not the least, SAGA filters characterize the particle bulk chemical components representative of the combined size range from the NOAA particle spectrometers.

**4 Conclusion**

The large range of conditions sampled by the high-quality aerosol instrument payload onboard the NASA DC-8 during the ATom missions provides unique opportunity to quantitatively investigate the comparability of submicron volume (and hence mass quantification) derived from physical sizing vs. bulk chemical instruments, as well as to evaluate whether currently reported AMS measurement uncertainties are realistic. Characterizing the upper end of the AMS transmission curve during field deployments is critical for meaningful intercomparisons. Calibrating the AMS transmission curve avoids improperly attributing the differences in transmission to errors in *CE* or *RIE* if a discrepancy is found. In-field calibration of AMS transmission is suggested since lens alignment or possible impacts during transport have been observed to cause a change in transmission. AMS variability in transmission can be significant, e.g., this study vs. Hu et al. (2017) and Liu et al. (2007), leading to differences of up to 25% in transmitted concentrations for ATom conditions, which could be larger in the presence of a larger accumulation mode. After applying the AMS transmission curve to the size spectrometer data, good agreement was found between the physically and chemically derived volumes over three orders-of-magnitude (slope = 0.96 and 1.09, $r^2$ = 0.95 and 0.93, for ATom-1 and -2, respectively). Significant deviations would have been observed if some literature transmission curves had been used. No evidence of biases in AMS detection of remote aerosols was found. The combined stated uncertainties are consistent for the overall statistics of the instrument comparison for the remote aerosols sampled during ATom.





The CU AMS inlet was equivalent to a $PM_{0.75}$ cyclone operating on ambient particles (i.e., not
dried prior to sampling) during ATom-1 to -3 and to a $PM_{0.95}$ cyclone during ATom-4. For an aerosol
density of 0.9 g cm$^{-3}$, such as pure hydrocarbon-like OA or cooking aerosol dominated by fatty acids, the
same AMS is equivalent to a $PM_{0.79}$ (ATom-1, -2, -3) and $PM_{1.0}$ (ATom-4) cyclone for dry particles.
Despite being equivalent to a $PM_{0.75}$ cyclone in ATom-1 and -2, 95 ± 15% of the theoretically calculated
URG $PM_1$ cyclone sampled mass/volume was detected by the AMS, as the effect of ambient pressure and
humidity on the URG cyclone transmission bridges the gap. Furthermore, the AMS quantified particle
mass and properties represent 68% (mean) of the integrated AMP volume and 41% of the integrated AMP
number from 2.7 nm to 4.8 µm geometric diameter ($d_p$) size range. PALMS-AMP at a 3-min time
resolution (or the PALMS-AMP products, which assumes a full coverage of >100 nm $d_p$ AMP)
characterizes 54% (76%) of the integrated volume and 5% (11%) of the integrated number, while MOUDI
1 µm stage impactor would collect 78% of the volume and 89% of the number. SAGA filters collect
nearly all the aerosol, 96% of the volume and 98% of the number. The more pressure-dependent cutoff
size of MOUDI or similar inlet that operates at ambient $P$ for airborne sampling may impact comparisons
with data from other instruments as a function of altitude. That effect could be compensated by lowering
the volumetric flow rate vs. altitude to keep the size cut (i.e., $d_{50}$) the same at the cost of a less sharp
transmission. The CU AMS inlet provides a more constant transmission vs. altitude. This work serves as
a case study of the importance of size ranges when intercomparing different instruments, and contributes
to document the performance of the ATom aerosol payload, confirms the realism of the stated
uncertainties, and serves as a framework for a subsequent intercomparison focusing on individual
chemical species.

***Acknowledgments.*** The authors gratefully acknowledge the support by NASA grants NNX15AH33A and
80NSSC19K0124. J.E.D. was supported by NASA grant NNX15AG62A. We thank the ATom leadership
team, science team, and the NASA DC-8 crew for their contributions to the success of the ATom mission.
We thank the AMS users community for many useful discussions, Bruce Anderson and Luke Ziemba for
collecting and sharing the LARGE particle extinction data during SEAC[4]RS, and Xiaoliang Wang and
Peter H. McMurry for inlet discussions and calculations. We thank Charles Brock, Christina Williamson,





and Agnieszka Kupc for the use of the AMP data, Joshua Schwarz and Joseph Katich for use of the SP2

data, and Karl Froyd and Daniel Murphy for useful discussions about PALMS.

*Data availability.* The ATom data is published at https://doi.org/10.3334/ORNLDAAC/1581.

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

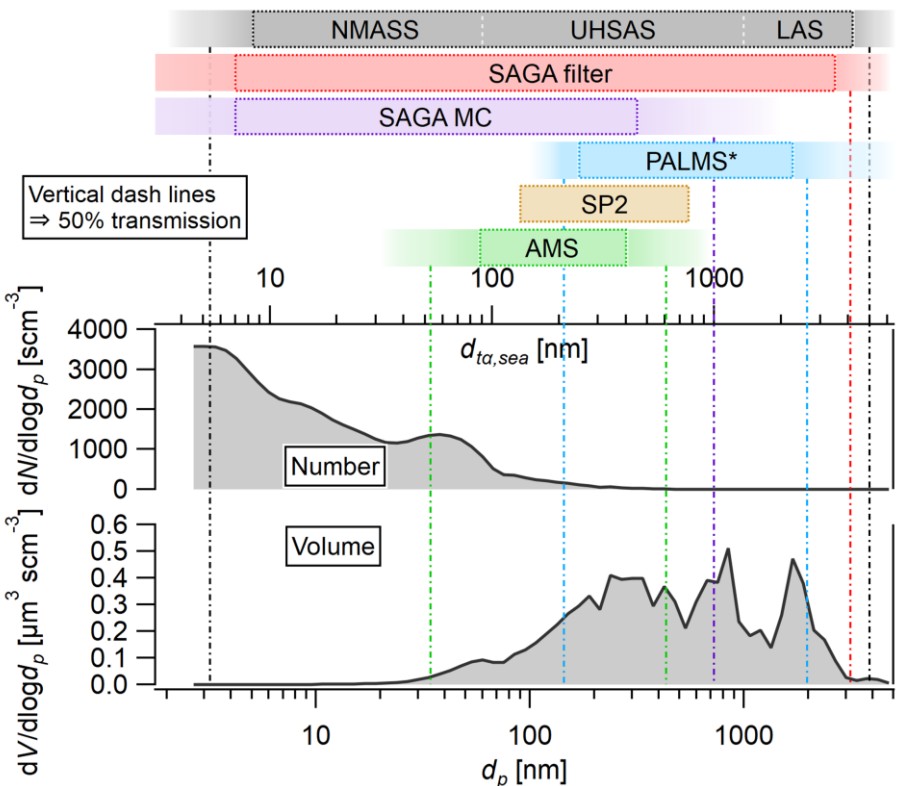

**Figure 1:** Approximate particle size ranges quantified by AMS, SP2, PALMS, SAGA MC, SAGA filters, and AMP (NMASS, UHSAS, and LAS), plotted with the campaign averaged AMP number and volume size distributions during ATom-2. For each instrument (except PALMS), the box indicates 100% inlet transmission and the transition shade on both sides indicates a decrease from 100% to 0%, with 50% denoted by the vertical dashed line. The PALMS bar represents the approximate size range contributing chemical information at a 60 min averaging time scale (at AMP size resolution) for composition data only (see Sect. 2.5). The top horizontal axis shows aerodynamic diameter ($d_{ta,sea}$) and the bottom geometric diameter ($d_p$); the conversion between the two diameters is based on ATom-2 campaign average aerosol density of 1.70 g cm$^{-3}$ and sea level $P$ of 1013 mbar using Eq. 2.

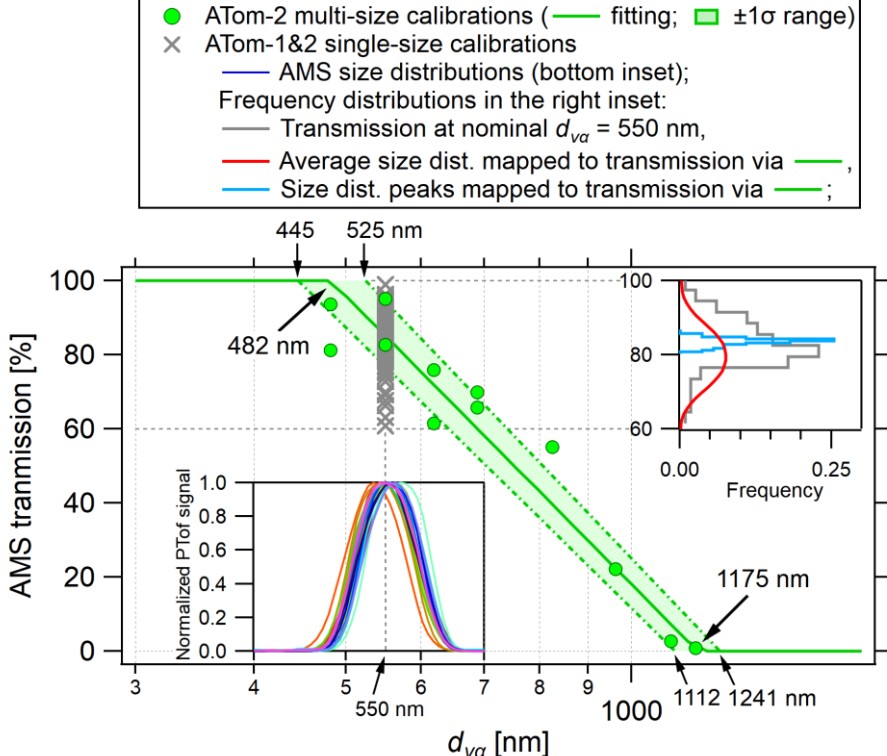

**Figure 2:** Results of in-field AMS large particle transmission calibrations vs. vacuum aerodynamic diameter ($d_{va}$) with $NH_4NO_3$ particles for ATom-1 and -2. The green markers are multi-size field calibrations, and the grey cross markers are single-size (at $d_m$ = 400 nm, equivalent to 550 nm $d_{va}$) field calibrations after every research flight. The insets show the frequency distributions of measured transmissions (right, top) and observed, normalized size distributions (left, bottom) of these single-size calibrations. A fit shows 100% transmission at 483 nm (1σ uncertainty of the fit: 445-525 nm) and 0% transmission at 1175 nm (1σ: 1112-1241 nm). When forcing 0% transmission at 1175 nm (confirmed by $(NH_4)_2SO_4$ calibrations), the fit to all data gives 100% transmission at 482 nm (1σ: 479-485 nm, not shown), consistent with the 483 nm inferred based only on the ATom-2 multi-size field calibrations.

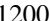

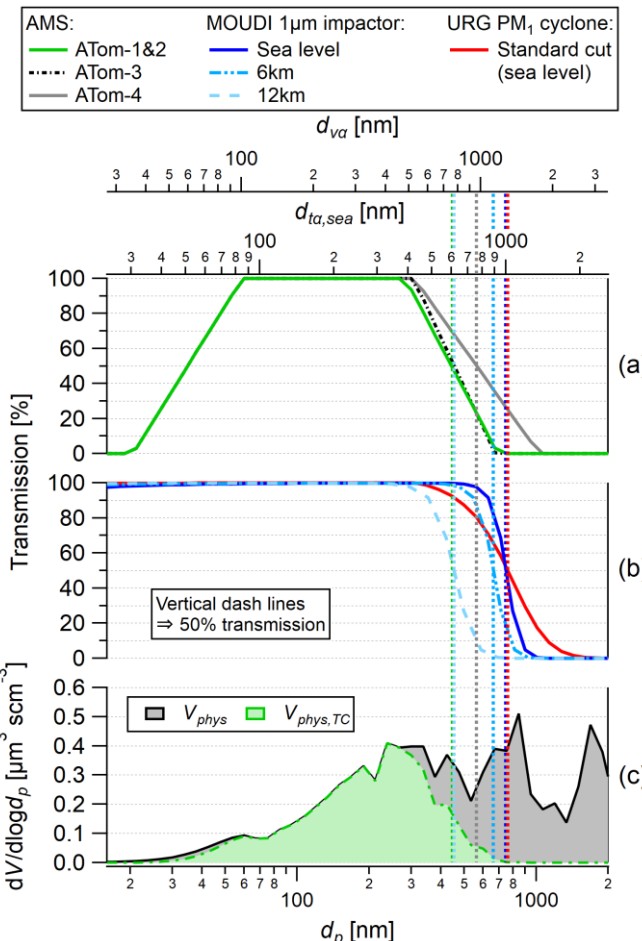

**Figure 3:** Transmission curves (a) for AMS during ATom-1 (same for ATom-2), -3 and -4 deployments, and (b) for MOUDI 1 μm stage impactor operated at sea level, 6 km, and 12 km (at $T = 293$ K as typical cabin temperature and $P$ based on the U.S. standard atmosphere) (NOAA, NASA, U. S. Air Force, 1976), and for URG $PM_1$ cyclone (sea level). (c) Average NOAA volume distribution ($V_{phys}$) and the fraction observed by AMS ($V_{phys,TC}$) for ATom-2. Particle size in geometric diameter ($d_p$; reported by NOAA), vacuum aerodynamic diameter ($d_{va}$; AMS), and aerodynamic diameter ($d_{ta}$; for the MOUDI impactor and URG cyclone; note that the MOUDI transmissions at 6 and 12 km are accurate on the $d_p$ and $d_{va}$ axes, but slightly off on the sea level $d_{ta}$ axis due to the change in slip correction) are shown as the three horizontal axes, all for dry particles. The 50% large particle cutoff sizes for AMS, MOUDI, and URG are listed in Table S1 based on $d_{ta}$, $d_{va}$, and $d_p$. Because URG cyclone is normally used to size-select ambient particles for ground studies, the equivalent dry cut would be smaller than 1 μm, approximately 0.8 μm based on the aerosols sampled in this study (Sect. 3.5).



**Figure 4:** Comparison between $V_{chem}$ and $V_{phys,TC}$ for (a) ATom-1 and (b) ATom-2, data points colored by altitude, and averaged to 5 min resolution. (c) A time series of the above two volumes as well as $V_{phys}$ for a research flight in ATom-2, with an inset showing the scatter plot only for this flight (at 1 min time scale, as well as for the time series). Note that $V_{chem}$ includes the AMS quantified sea salt. Two correlations coefficients ($r^2$) are listed: one at linear scale (commonly used) and the other at logarithmic scale, which emphasizes the scatter at low concentrations.

**Figure 5:** Box plots of $V_{chem}/V_{phys,TC}$, and the linear regression fitting slopes and correlations of the two volumes for (a) the combined ATom-1 and -2 data sets, (b) ATom-1, (c) ATom-2, binned by removed $V_{phys}$ fraction when applying AMS transmission (at 20% interval). 10th, 25th, 50th, 75th, and 90th percentiles are plotted with the box and whiskers. The binned scatter plots can be found in SI as Fig. S20. (d-i) are the normalized frequency distributions of the volume ratio for ATom-1 and -2, respectively, at three averaging time scales: (left) 1 min, (middle) 5 min, and (right) 10 min. The green-tinted backgrounds indicate the combined 2σ accuracy from AMS (38%; 2σ) (Bahreini et al., 2009) and UHSAS (+12.4/-27.5%; treated as 1.5σ in this study) (Kupc et al., 2018).



**Figure 6:** Sensitivity test of AMS transmission: the regression slopes and correlations between $V_{chem}$ (y-axis) and $V_{phys,TC}$ (x-axis) by artificially changing the AMS transmission. The four subpanels labeled with (b), (c), (d), and (e) are for the four anchoring points, (i) 35 nm, (ii) 100 nm, (iii) 482 nm, and (iv) 1175 nm (all in $d_{va}$), as shown in the top AMS transmission figure. In (d) and (e), the green-tinted background indicates the one standard deviation range from in-field calibrations, and the orange-tinted background in (d) is the narrower standard deviation range estimated from multiple calibrations (Fig. 2).



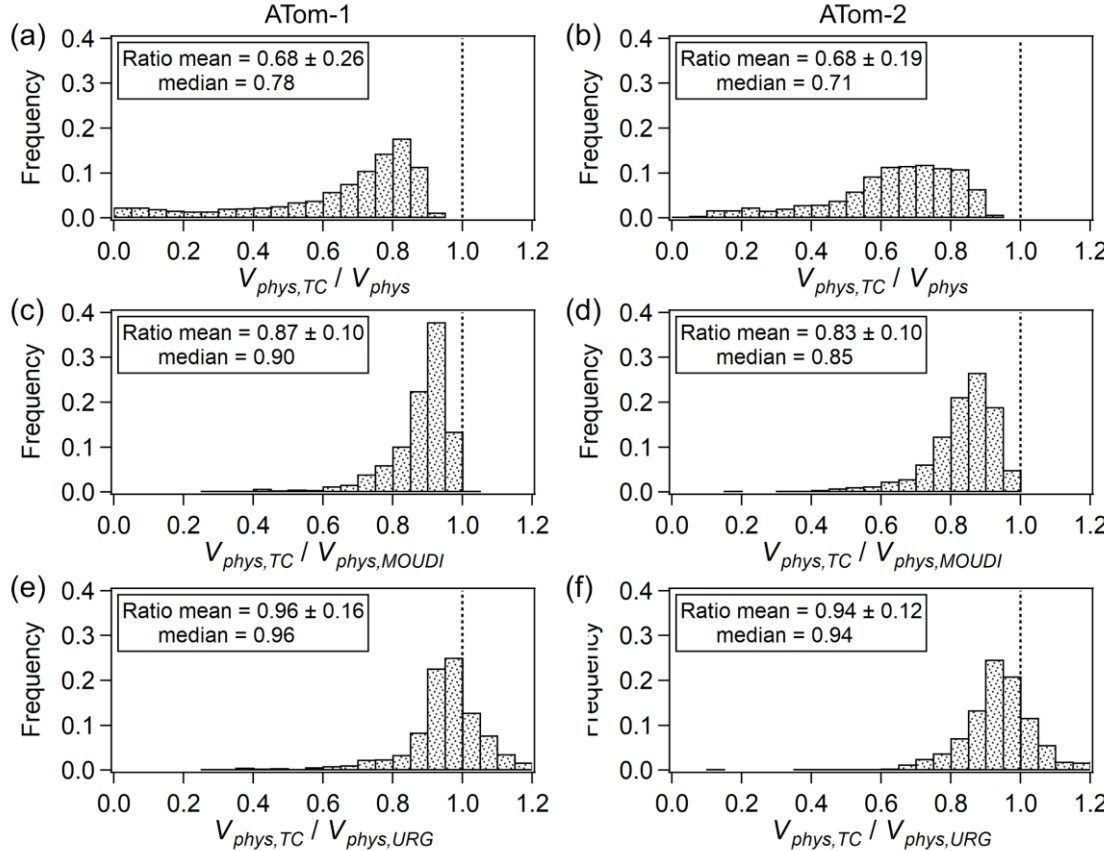

**Figure 7:** (a-b) Frequency distributions of the AMS-transmission-corrected $V_{phys}$ ($V_{phys,TC}$) vs. the (total)
$V_{phys}$. (c-d) Same for $V_{phys,TC}$ vs. the dry condition altitude-dependent MOUDI-1μm-stage-impactor-
transmission-corrected $V_{phys}$ ($V_{phys,MOUDI}$). (e-f) Same for $V_{phys,TC}$ vs. the ground level ambient condition
URG-PM$_1$-corrected (standard PM$_1$ cut) $V_{phys}$ ($V_{phys,URG}$). ATom-1 is shown on the left and ATom-2 on
the right.





**Figure 8:** Campaign-averaged volume (left) and number (right) size distributions observed by AMP in ATom-2 (NMASS measured down to 3 nm and here we only show the subrange starting from 8 nm), together with the approximate particle size ranges contributing chemical composition information (without consideration of the details of the chemical detection) to the AMS, PALMS, and SAGA filter, and size-selected by a MOUDI 1 μm stage impactor. The top panel is one dimensional with the campaign average result of each instrument (the transmissions of MOUDI and SAGA filter are altitude dependent and plotted in Fig. 3 and Fig. S18, respectively; PALMS effective detection range depends on counting statistics, and the detected particles given a sampling period are discussed in Fig. S14-15). Note that the top panel shows the fraction of the average, while Fig. 7 shows the average fractions (a summary at Table S2). The right plots represent the size ranges of the number size distribution contributing chemical information to each instrument. The following panels show the vertical profiles of the same quantities for AMP, SAGA filter, MOUDI impactor, AMS, and PALMS-AMP, respectively. The PALMS-AMP product (Froyd et al., 2019) reports composition above 100 nm, the size range indicated by the dashed square in the bottom panels. The plotted altitude bins are 800 m each.



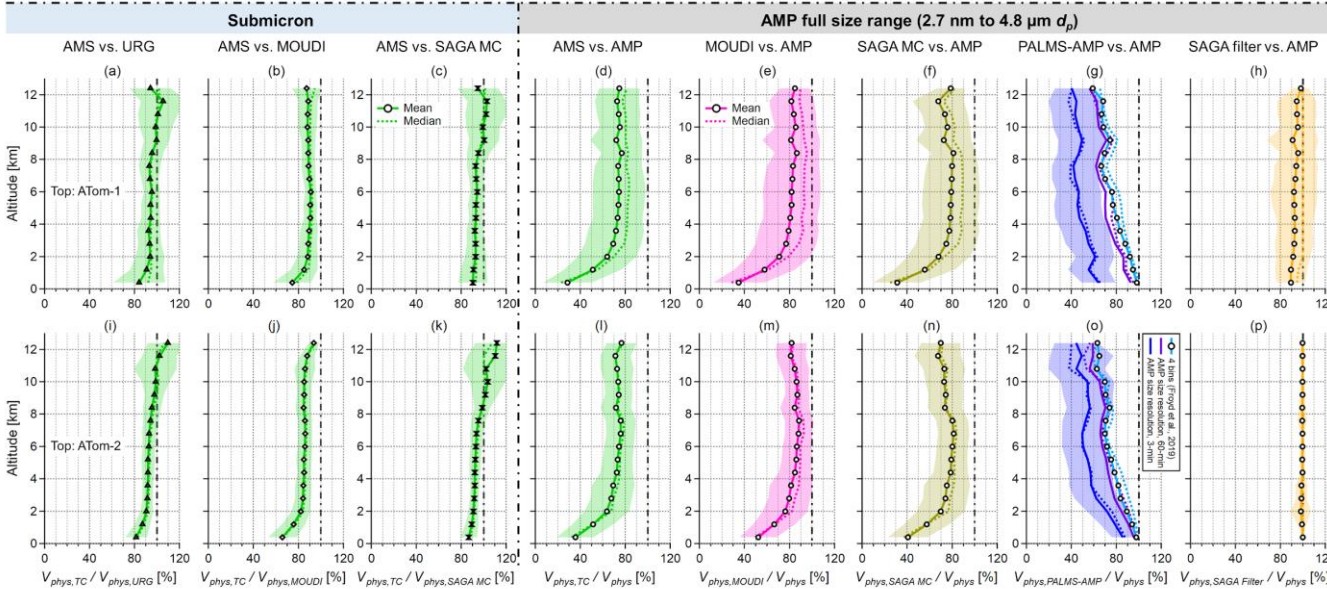

**Figure 9:** Comparison of the fraction of the particle volume that is observable (i.e., those contributing chemical composition information, but independent of the properties of the chemical detector) between instruments or inlets as a function of altitude, for the conditions in (top) ATom-1 and (bottom) ATom-2. On the left, the widely used approximate submicron cuts are compared. On the right, the ATom aerosol payload is compared, including a MOUDI 1µm impactor that has been flown in other studies. The color-shaded area indicates the SD of volume ratios.