# Peer review of "The Importance of Size Ranges in Aerosol Instrument Intercomparisons: A Case Study for the"

_Atmospheric Measurement Techniques, 2020_

## Short Comment (SC1) · 27 Jun 2020

Comment on Guo et al., The Importance of size ranges in aerosol instrument intercomparisons: A case study for the ATom Mission

D. M Murphy1, K. D. Froyd1,2, C. A. Brock1, C. Williamson1,2, and A. Kupc1,3

1Chemical Sciences Laboratory, National Oceanic and Atmospheric Administration 2Cooperative Institute for Research in the Environmental Sciences, University of Colorado 3Faculty of Physics, Aerosol Physics and Environmental Physics, University of Vienna, Austria

[Figure]

**Introduction**

The authors of this comment are the principal investigators for the PALMS single particle mass spectrometer and the aerosol microphysical properties (AMP) size distribution instruments that were operated on the Atmospheric Tomography Mission (ATom). The PALMS authors are the developers of the instrument and associated data analysis methodologies, and have applied the instrument to study the atmospheric aerosol for more than 2 decades.

We write out of concern that this manuscript (Guo et al., 2020; hereafter G20) has serious flaws in the way it represents the PALMS instrument capabilities and the PALMS+AMP data products submitted to the ATom database. Others who use data from ATom and other missions could misunderstand how to utilize PALMS+AMP products in the mission archive. Moreover, G20 introduces into the literature a new "chemical information" metric that has unphysical behavior and lacks mathematical rigor.

This comment will show that that G20: - Analyzes the PALMS data with arbitrary size bins that are too narrow. The combination of a very high size resolution and prescribing zero values to any bins without PALMS particle observations biases the results. - Includes a conceptually flawed "chemical information" metric that leads incorrect conclusions. - Improperly compares instrument performance by using different metrics for different instruments. - Is quantitatively incorrect in several ways: underestimating by large factors the number of particles actually sampled by PALMS, and producing concentrations that are biased low and scale unphysically with sampling time.

These issues were pointed out to the lead authors, who are PIs and scientists associated with the Aerodyne Aerosol Mass Spectrometer (AMS) instrument operated on ATom before submission. There was extensive, but unresolved, discussion, that included us suggesting alternative figures and text. We repeatedly stated that the method of calculating PALMS statistics was not consistent with a detailed description of PALMS data analysis methods (Froyd et al. 2019; hereafter F19) and therefore was likely to

be inaccurate, and that their "chemical information" metric was both conceptually and quantitatively flawed. Because of the use of PALMS data, K. Froyd and D. Murphy were originally invited to be coauthors on this manuscript but we were removed as authors when agreement could not be reached. Following this, four other authors (comprising the AMP team plus one other) removed themselves from authorship.

The purpose of this comment is to describe these issues in detail as a matter of public record, to help mitigate any misconceptions about the intrinsic merit and capabilities of the PALMS instrument and associated data products, and to form recommendations for the Editor to correct these problems before acceptance of G20 in Atmospheric Measurement Techniques. While much of G20 represents solid research and a valuable contribution to the literature, the sections relating to PALMS are flawed and need to be removed or undergo major revision, as detailed at the end of this comment. The PALMS-AMP data products are very complementary to AMS and SAGA. Each measurement has its own strengths. We continue to be ready to assist the AMS team in representing our data.

Background

PALMS by itself cannot derive absolute particle concentrations, as it samples particles with uneven efficiency across its nominal size range of 0.1 to 4.8 $\mu$m. This uneven size sampling underlies the methodology developed by F19 which derives absolute concentrations from PALMS by combining it with accurate counting and sizing instruments (the AMP size spectrometers for ATom). Members of the AMS team are coauthors on F19.

Arbitrary and narrow size bins

G20 show results obtained by combining PALMS and AMP data (e.g., Fig 8, 9, S28, S29, S30). However, they explicitly do not follow the data analysis procedures used by the PALMS+AMP team for data submitted to the mission archive. Instead, the manuscript "provides an alternative illustration of PALMS size coverage" (line 339).

Instead of the 4 size bins used by the PALMS+AMP team to derive absolute concentrations, G20 use 20 bins per decade of diameter, or 35 bins across the nominal PALMS size range. The choice of bin width is critical because it significantly affects their analysis of data coverage and derived concentrations. The G20 choice has no physical basis. Presumably, it arises because the AMP team uses 20 bins per decade as a convenient bin size to report optical particle counter data in the data files, minimizing file size while providing adequate resolution for optical calculations. Note that 20 bins/decade is narrow: each bin is only a ~12% change in diameter. When establishing the methodology to derive concentrations from PALMS, F19 found that grouping the size-resolved PALMS composition data into just four size bins was a good compromise between adequate statistics and changes in particle composition with size. This choice of bin width is extensively and quantitatively evaluated in F19, which states "It is infeasible to retain the raw size resolution of the OPS [optical particle spectrometer, AMP here] for the integrated concentration analysis".

The G20 manuscript creates a data coverage metric called "chemical information content" or "chemical composition information". The metric, used in G20 to derive Figs. 8, S29, and elsewhere, assigns an information content of 1 to size bins with particles and 0 to size bins without particles. It is easy to see that this leads to lower values of "chemical information" as the number of size bins is increased.

Imagine, for example, that an instrument has measured 50 particles. If there is no size resolution (one bin), the "chemical information" is always 1. If there are three bins and 50 particles, statistically each bin will almost always contain a particle and the average "chemical information" will be very close to one. But if there are 100 size bins (e.g. the native resolution of the UHSAS optical spectrometer) and 50 particles, at least 50 bins are necessarily empty and the average "information content" will be less than 0.5. With 1000 bins, in this example the "chemical information" metric that is supposed to quantify data coverage would be reduced to <0.05. Although of course G20 are not proposing 1000 bins, any metric that can produce arbitrarily small values of the supposed data

coverage simply by choosing narrower bins must be mathematically invalid.

Conceptual flaws in the "chemical information" metric

It is worth exploring some reasons why their metric of data coverage has an unphysical dependence on the width of the size bins. Even if reasonable bin resolution were used and the quantitative errors described below were corrected, the "chemical information" analysis is still conceptually flawed as an indicator of data coverage.

The first reason the "chemical information" metric in G20 is flawed is that it penalizes high-resolution data for empty bins but does not give credit for information obtained from narrower bins. For example, having two size bins instead of one adds information about possible differences in composition between small and large particles; G20 assign no value, either conceptually or mathematically, to this added information.

Second, the G20 analysis does not recognize autocorrelation. In reality, neighboring bins are not independent of each other, either in concentration or chemistry. This is particularly true at the high resolution used by G20. For example, if 200 nm particles are composed of 80% sulfate, it is very unlikely that 225 nm particles are composed of pure organics. If 200 nm particles truly had a completely different composition than 205 nm particles, and those again different than 210 nm particles, etc., PALMS would need to sample an enormous number of particles to fully characterize the size-dependent composition. If all the particles from 100 nm to 5 $\mu$m in a given air mass were exactly the same, PALMS would only need to sample a few particles. Naturally, the amount of data required is in between these extremes.

When analyzing PALMS data coverage, G20 do not consider that different size particles may have correlated compositions, yet that assumption is implicit in the interpretation of AMS and SAGA data. Without it one would have no idea, for example, if a 50:50 organic-sulfate mixture represented mixed particles, small pure organic particles and large pure sulfate particles, or vice versa. The AMS collection efficiency (bounce) correction, which is based on bulk composition, also assumes similar particle compo-

sitions at different sizes. For example, if 150 nm particles were ammonium sulfate and 250 nm particles were sulfuric acid, the AMS bounce correction would be incorrect.

A rigorous analysis of information and data coverage is beyond the scope of this comment but we can point out that that it should be framed in terms of detection limits. The G20 "chemical information" metric conflates information about the instrument with information about the atmosphere. In a typical 3-minute period in the stratosphere PALMS measures no 30 nm particles and no 3 $\mu$m particles. G20 assign the same zero value to these, yet they are very different. The former is a statement about PALMS because 30 nm is far below PALMS' nominal size range, whereas the latter is a statement about low concentrations in the stratosphere. In G20 this is evident in Fig. 8 for example, where zero "information content" is assigned at high altitudes to large sizes that PALMS samples well (because the atmosphere there contained almost no large particles). This illustrates why the scientific literature conventionally discusses detection limits for in-situ instruments rather than "chemical information". When one compares an instrument detection limit with an atmospheric concentration it properly separates instrument performance from atmospheric properties. There are mathematically rigorous ways of defining information content for aerosol size distributions (Preining, 1972) in the context of a priori information (Petty, 2018), yet an arbitrary metric is developed in G20 instead.

Flawed instrument comparisons

The G20 manuscript applies different metrics to different instruments, leading to biased comparisons. In particular, G20 Fig. 8, the primary comparison of instrument measurement capabilities, is not internally consistent. The "chemical information" metric is applied only to PALMS, not the other instruments. PALMS data coverage is derived using a limited (3 minute) sample time. Imposing this sample time restriction only on PALMS misrepresents its size range and data coverage relative to the other instruments. Were a similar information content or sample time restriction imposed on AMS and SAGA, a high fraction of samples would be below detection limit. At the native sampling times

for AMS or SAGA (1 min and ∼5-15 min, respectively), 72% of AMS and 35% of SAGA samples are below their detection limits for a major chemical component (sulfate, ammonium, nitrate, or organic material). Such samples have little "chemical information" across all sizes beyond being able to say "below detection limit", yet that is ignored in G20 Fig 8, where instead AMS and SAGA are shown to detect 100% of particles within their nominal size ranges. On the same 3 minute sample time as PALMS, over 60% of AMS organic data in the tropical Pacific are below detection limit, yet G20 assert that those data have more "chemical information" than PALMS data above detection limit.

Quantitative errors: scaling and number of particles.

A quantitative error in G20 is that their derived PALMS+AMP concentrations in Fig. 8 and elsewhere scale improperly with sampling time. This is illustrated in Fig. S29, where increasing the PALMS sampling time from 3 to 60 min increases the PALMS+AMP number and volume. Although PALMS observes more particles with longer sample time, higher particle counts do not translate into higher derived concentrations. The G20 method to derive PALMS+AMP concentrations is therefore flawed: physical concentrations do not scale with sample time. The method of F19 does not have this problem.

Figure 1 below shows that it is essential to use information across wider bins than 20/decade to obtain accurate concentration measurements with PALMS+AMP. If one simply puts zeros in empty bins, as in G20 Fig. 8, the resulting concentrations have large discrepancies and are biased low.

The G20 manuscript also strongly underestimates the number of particles sampled by PALMS. Fig. S14 in G20 is the basis for the derived PALMS+AMP concentrations in Fig 8 and elsewhere. Although their estimated sampling rate near the peak sampling efficiency at around 300 nm appears reasonable, their estimates have large errors for smaller and larger particle sizes, exactly those most important for data coverage (Figure 2). For instance, for the narrow bin at 1005-1128 nm G20 calculate that PALMS

observed ~500 particles over an ATom deployment, whereas PALMS actually observed over 14000. Considering the 5 bins/decade curve on G20 Fig. S14, they underestimate the average sampling rate for the bin centered near 130 nm by a factor of about 10 and underestimate the PALMS sampling rate above 1 $\mu$m by factors of 8 to 40. Similar errors apply to all other curves in S14. These errors presumably propagate into all calculations in the manuscript (Fig 8, 9, S15, S28, S29, S30, and Table S2). Underestimating the number of particles sampled by PALMS by x10 and higher factors will seriously affect the conclusions of the manuscript.

Specifically, G20 state (line 345) "The probability of detecting on average one valid particle per AMP size bin in the PALMS is very low below ~160 nm and above 1000 nm over a typical 3 min analysis period." In contrast, on average PALMS actually measured in 3 minutes the composition of about 4 particles smaller than 160 nm and about 10 particles larger than 1 $\mu$m. G20 (lines 336 and 694) also refer to PALMS+AMP products for small and large particles as "extrapolations". In reality, during the ATom deployments PALMS measured over 100,000 particles between 100 and 180 nm diameter and almost 100,000 between 1 and 4 $\mu$m.

We provided the AMS team with unpublished data on the PALMS sampling rate (the red curve on their Fig. S13), and allowed them to use these data even after we were removed as coauthors. We have been unable to replicate exactly how they arrived at their underestimates of PALMS sampling rates. The manuscript states (Fig. S14 caption) that they used a detection efficiency and multiplied it by a flow rate and atmospheric concentrations. However, according to F19, using a detection efficiency curve for PALMS is "not recommended due to many possible pitfalls and large, unquantifiable errors."

Summary

The incorrect calculation of the number of particles sampled by PALMS and the use of arbitrarily narrow bin widths together lead to low assessments of the PALMS data

coverage. G20 assert (line 691, also Table S2 and Figs 8 and 9) that the PALM+AMP data characterize about 54% of the aerosol volume within 3 minutes of sample time during the ATom flights. Figure 3 shows cumulative size distributions for the ATom flights. One can see that a 54% fraction of the volume within the nominal PALMS size range is implausibly low.

The analysis of the PALMS+AMP data in G20 has both quantitative and conceptual errors. Even if the quantitative errors were fixed, the "chemical information" metric of data coverage would still be conceptually flawed. G20 Figs. 8 and 9 and the stated size-dependent numbers of PALMS-sampled particles are incorrect by large margins, as is the 54% PALMS volume coverage figure. Finally, instruments are not compared using the same criteria: only the PALMS data are scaled by "chemical information" using a limited sample time, and an implicit assumption that particle composition is uncorrelated across nearby diameters is applied only to PALMS.

The manuscript is not suitable for publication unless the incorrect calculations, the "chemical information" analysis, and all associated discussion are removed. Specifically, in G20, Figs. 8, 9, S14, S15, S28, S29, and S30 show incorrect information about PALMS. The associated discussion starting on line 331 is incorrect, as are the data coverage percentages following line 689, line 744, and in Table S2. The red curve on Fig. S13 is correct but mislabeled.

Instead of the existing panels showing PALMS+AMP data coverage in Fig. 8, we would support a figure similar to the bottom panel of G20 Fig. S29 or S30. These do a good job of illustrating the coverage of the PALMS data and how PALMS samples within the four broad size bins established in F19.

References

Froyd, Karl D., Daniel M. Murphy, Charles A. Brock, Pedro Campuzano-Jost, Jack E. Dibb, Jose-Luis Jimenez, Agnieszka Kupc, Ann M. Middlebrook, Gregory P. Schill, Kenneth L. Thornhill, Christina J. Williamson, James C. Wilson, and Luke D. Ziemba,

[Figure]

A new method to quantify mineral dust and other aerosol species from aircraft platforms using single-particle mass spectrometry, Atmos. Meas. Technol., 12, 6209-8239, https://doi.org/10.5194/amt-12-6209-2019, 2019.

Guo, H., P. Campuzano-Jost, B. A. Nault, D. A. Day, J. C. Schroder, J. E. Dibb, M. Dollner, B. Weinzierl, and J. L. Jimenez, The importance of size ranges in aerosol instrument intercomparisons: A case study for the ATom mission, Atmos. Meas. Tech. Disc., submitted manuscript, 2020.

Petty, G. W., On some shortcomings of Shannon entropy as a measure of information content in indirect measurements of continuous variables, J. Atmos. Oceanic Technol., 35, 1011-1021, https://doi.org/10.1175/JTECH-D-17-0056.1, 2018.

Preining, O., Information theory applied to the acquisition of size distributions, J. Aerosol Sci. 3, 289-296, https://doi.org/10.1016/0021-8502(72)90050-X, 1972.

Figure 1. PALMS+AMP sulfate mass concentrations if one inserts zero values in bins without particles (20 bins/decade) in a 3-minute period, compared to the PALMS standard method used for data in the ATom mission archive (x axis).

Figure 2. A comparison of the G20 calculation of the number of particles ostensibly sampled by PALMS with the number actually sampled. The red dashed curve is from Fig. S14 in G20. The black curve is the average number actually sampled out of cloud expressed as particles per 3 minutes. This shows 5 bins/decade; other bin sizes have similar discrepancies. Thin dashed lines are individual deployments. (Slight changes in vacuum inlet alignment favored large particles during ATom 2 and 3 and small particles during ATom 1 and 4.)

Figure 3. Cumulative size distributions expressed as the fraction of aerosol volume below 3 $\mu$m optical diameter. Solid lines are data taken below 2 km altitude and dashed lines above 2 km. Horizontal grey bars show approximate size ranges for AMS and PALMS+AMP data. These are averages of all ATom data out of cloud and west of 60°

W longitude. The last criterion is just a simple way to exclude strong biomass and dust plumes near Africa.

[Figure]

**Absolute concentrations
of Sulfate Mass (ug/std-m3)**

AMS analysis of PALMS compared
with standard PALMS method

For AMS method, diambins with
"no PALMS chemical content"
contribute zero mass

(Y-axis) AMS analysis of PALMS

(X-axis) PALMS standard method (ATom data products)

**Fig. 1.**

**Fig. 2.**

Number of PALMS particles in 3 minutes with 5 bins/decade

Optical diameter (µm)

- Guo et al. Figure S14
- Actual PALMS average
Thin lines: ATom1 to 4

[Figure]

**Fig. 3.**

---

## Referee Comment (RC1) · Anonymous Referee #1 · 16 Jul 2020

General Comments: The manuscript by Guo et al. provided a critical evaluation of the size-related factors impacting aerosol intercomparisons and aerosol quantification during NASA Atmospheric Tomography Mission (ATom), with a focus on the Aerosol Mass Spectrometer. This is an important and necessary piece of work, which can fill the absence of significant unknown biases and the appropriateness of the accuracy estimates for AMS total mass/volume for the mostly aged air masses encountered in ATom. Overall, the paper is well-written. One of the major concerns of this study, however, is the reason for choosing certain bin width, which influences particle size distribution and subsequent comparisons. Before its publication, the following comments need to be addressed.

1.The reason for the choice of bin width (line 337) needs to be elaborated. What is the difference of size distributions in different seasons between using this bin width and using others (narrower or broader width)? The authors need to address such uncertainties in the revised manuscript. 2.The manuscript focused on size-related intercomparisons, but what are the uncertainties in using the same CE for all particle sizes? 3.The PALM+AMP characterize about 54% of Vphys, which is much lower than that in ATom-1 and -2, the SAGA filters, MOUDI, AMS (line 689-691). Please elaborate. 4."r" in "r2" should be in italics.

Please also note the supplement to this comment: https://amt.copernicus.org/preprints/amt-2020-224/amt-2020-224-RC1-supplement.pdf

---

## Referee Comment (RC2) · Anonymous Referee #2 · 22 Jul 2020

The paper from Guo et al. reports on the importance of the size ranges of both inlets and instruments in aerosol intercomparison with a special focus on airborne measurements during ATom Mission. Such a systematic analysis is essential for better understanding the difference that can be observed when comparing different instruments or inlets to each other, especially during aircraft measurements where altitude can strongly affect the size range of the aerosol inlet and/or instruments. This work is, therefore, suitable for a publication to AMT. However, several issues first must be clarified.

Major comments:

- First of all, the authors must reexamine what is the real focus of the manuscript. Although the title mentions "size range in aerosol instrument intercomparison", abstract and introduction only refer to AMS issues, while only a few words are made on the other inlets and instruments on the conclusion. This does not reflect both manuscript and supplementary information, which are deeply discussing the effect of altitude on the size-range of different inlets and instruments without directly comparing them to AMS results. Consequently, if the manuscript aimed to investigate the effect of the AMS aerodynamic lenses transmission efficiency during intercomparison exercises during ATom Missions, discussions on MOUDI-PM1, URG-PM1 inlets, SAGA and PALM-AMP are not needed on the manuscript since the AMS was not connected to such inlet or no direct compared to them. On the other hand, if the target of the authors was to discuss how the size-range of inlets and individual instruments are affected by altitude during the ATom Missions, then the authors may need to reconsider the discussion on the AMS and the comparison Vchem vs. Vphys or Vphys-TC.

- According to the title, I would expect more direct intercomparison between the different results providing by the instruments, similarly than for the AMS (Vchem vs Vphys-TC) rather than simply comparing numbers or transmission profiles. For example, why the authors did not compare their AMS results with SAGA, and PALMS-AMP? how the change in transmission efficiency of each instrument affect the comparison?

- Only very spare information is done on the different methods used to estimate the change of the size-range of the instrument with the altitude. It would be great to have, for example, a dedicated section on it detailing the different hypotheses and models used for the calculation.

- Focusing on the AMS, the authors did not convince me of the importance of transmission efficiency. Although the focus of the manuscript (based on abstract and conclusion) is to demonstrate the importance of the transmission efficiency of the aerodynamic lenses of the AMS during intercomparison, it is first mandatory to ensure, before any comparison that all the reference instruments are properly calibrated. Here,

the calibration and quality assurance of the different optical size spectrometers are not mentioned. How do the 3 instruments compare on their overlapping size-range? The manuscript will certainly gain consistency if the authors can also discuss these points as well as their uncertainties rather than only citing literature (lines 301-307) similarly than for the AMS. In such an exercise, not only the AMS had to be carefully calibrated.

- Another critical aspect is the absence of discussion when comparing correlations with and without applying the size-dependent correction method. Figure S4 shows a very good correlation between the instruments without applying it. The discussion on the effect or not of the size-dependent correction should be supported by quantitative numbers as well to demonstrate that transmission efficiency of the AMS plays an important role (which was up to now never considered on the literature) and properly quantify how much Vchem can be affected by the transmission efficiency of the AMS lenses. The reason is certainly on the definition of the Vphys itself. Why Vphys has a volume concentration ranging up to 4.8 $\mu$m? This is quite surprising since there is a long discussion on the change of the size-range of the AMS between the missions (section 3.1 and 3.2). Moreover, the large fraction of super-micrometer particles on the volume size distribution is controlling most of the total volume mass concentration on Vphys. Why the authors did not cut Vphys to 1 $\mu$m for all missions? Or more simply using Vphys-0.75 or Vphys-0.95, corresponding to their own estimated AMS upper-size-cutting (PM0.75 for ATom 2 and PM0.95 for ATom2, for example). It would also be possible to consider a volume size distribution up to 800 nm (equivalent diameter similarly to most of the SMPS system). Applying such kind of approach provides i-/ a comparable size-range between the 2 instruments and ii-/ the possibility to extrapolate the results to most of the AMS work that was done up to now, which are mostly using SMPS for comparison. Only then, the authors can properly discuss the importance of considering or not the aerodynamic lens transmission efficiency on the comparison with collocated measurements, and how much it can influence the correlation. It would be also interesting to do a sensitivity test based on the influence of the coarse mode concentration (low and high coarse mode periods). For this purpose, Table S2 is certainly one of the most

important for the comparison between the different instruments. However, I don't fully understand why only Vphys-TC is used and not a systematic Vchem and Vphys-TC on a similar size range to quantify the effect of the AMS transmission efficiency correction on different systems.

- Does it make sense to apply the transmission curve on Vphys? This approach makes Vchem and Vphys dependent variables. Furthermore, assuming the size-spectrometers have perfect transmission efficiency and particle detection up to 4.8 $\mu$m, it is not completely surprising that after applying the AMS transmission efficiency correction, a perfect correlation is obtained. Finally, it does not say that the two instruments better correlate than without applying the correction. Instead, it would be better to correct the AMS mass or volume concentrations by the corresponding transmission efficiency. This means first ensure that ePTOF and MS mass concentrations agree, then correct the ePTOF mass concentration from transmission efficiency to get the final AMS mass concentration. Then, it would be possible to compare to parallel measurements of the same size cutting. However, I have to recognize that I am not sure whether this is feasible on this dataset or not.

- Although, the authors carefully investigate the transmission efficiency of their instrument; key information on the calibration protocol itself is not mentioned. For example, how the particles were generated and selected (especially for mono-dispersed super-$\mu$m ammonium nitrate particles)? Which instrument was used as a reference in parallel to the AMS? The authors used 400 nm (mobility) ammonium nitrate particles to performed their IE calibrations, however, looking at Figure 2, this diameter seems to be already out of the 100 % transmission efficiency of their aerodynamic lenses having rather a transmission ranging from 80-95%. How did the authors deal with that? Does it mean that their RIE (and the resulting AMS mass concentration) is underestimated?

- Last but not least, the reported discussion regarding PALMS and PALMS-AMP analysis must be clarified and solved on the revised version of the manuscript.

Minor comments:

Abstract:

- The abstract is rather unspecific and gave poor information on the "critical evaluation of the size-related factors", the "good agreement" when comparing AMS with "size spectrometers" and "standard PM1 volume". Without any indication of the transmission efficiency, the size range of the reference used, and results before and after applying the correction, how can we conclude from an "absence of significant unknown biases"? More details on the "upper end of the AMS size-dependent transmission" as well as the upper size cutting of the size-spectrometer must be provided here.

- Why results from the other instruments (SAGA, PALM) and inlets (MODI-PM1 and URG-PM1) are not mentioned in the abstract?

- line 28: What is a good agreement? Please provide number

- line 29: Please precise from which size-spectrometer (and its size-range) were used.

- line 31: What is a "standard PM1 volume"?

Introduction: Over the last two decades, there is a lot of works reporting comparison between AMS and collocated instruments either for individual chemical species and/or total mass/volume. Most of the time the correlations are within the different instrumental uncertainties. Therefore, the authors should provide more clear justifications for their motivations and the need for such correction on the introduction.

- line 53: ACTRIS and not ACTRiS

- line 65: Could you clarify the sentence "Volume comparisons probe the ability of the AMS to quantify total mass and predict aerosol density based on fractional composition accurately, and hence it the most germane comparison for total quantification"

- line 73-75: Does it make sense to repeat this information. This was already mentioned in the introduction.

- line 112: Details on the modifications made on the AMS compared to a "standard" one would be appreciated.

- line 112 – 284: I think a detailed description of the AMS is not needed. AMS is not a new instrument and it was already described in the cited papers. Moreover, I don't follow how section 2.3 is splitting. For example, the authors first discuss the factor influencing the collection efficiency (line 125-145) and then with the impact of the RH and the CDCE (line 194-199). A similar comment can be made for the IE and RIE. The current splitting leads to unnecessary repetitions. For example line 143-144: "CE is estimated to contribute substantially to the overall uncertainty of the AMS concentration measurements (Bahreini et al., 200), line 169-170 "For the AMS reported mass concentration, uncertainties (i.e. accuracies) in CE (30%)", and line 211 "[. . .] are similar to those from Bahreini et al. (2009), due to the dominate uncertainties contribution from CE (30%)". Similarly, the uncertainties of the RIE is discussed on line 157-172 and again online 204-211. A single AMS section will improve it.

- line 269: density calculation (eq. 5): why not directly include sea-salt since i-/ you quantified it on your AMS measurements (line 256) and ii-/ you include it anyway later (line 382-386). A similar comment can be made for rBC. Then you can discuss once how total mass concentration and density were estimated. Here again, it will improve the clarity of the manuscript.

- line 272: Density of OA was calculated based on the Elemental Analysis of the organic mass spectra ranging from 1.51 – 1.59 g cm-3 (line 280) but the default OA density is set to 1.7 g cm-3. Does the default value make sense? Could it be that the value of 1.7 is artificially high due to low concentration and overall strong contribution of CO2+ to the total mass spectra?

- line 301: Vphys is a combination of 3 optical particle size-spectrometer ranging from 2.7 nm to 4.8 $\mu$m. How these 3 instruments are comparing together on their overlapping size range?

- line 358: Do you need to include the SAGA on the manuscript since it is no use for the comparison with the AMS?

- line 441: Is this section necessary, since none of the instruments used for the discussion were connected to such inlets?

- line 515: the two following sentences are unclear and need to be rephrased "AMS observes a fraction of Vphys (Figs. 1 & 3). AMS transmission vs. dp was calculated based on the calibrated transmission vs. dva (Eq. 1) and the AMS estimated time-dependent pm (Eq. 5), and applied to the AMP size distributions used to derive Vphys."

- line 520: How is the correlation without corrected Vphys by AMS transmission?

- line 523: "slightly worse"? Could we say that this correlation is slightly worse according to all instrumental uncertainties? Both are within 10%. Instrumental uncertainties on the comparison should be discussed. Which type of regression was used (classical linear fit or least orthogonal distance fit)?

- Figure 1: Please use a log-scale for dN/dlogd to better see the particle number > 100 nm.

- Figure 4: Can the strong increase of Vphys compare to Vchem at the lowest altitude simply be associated with marine sea-salt coarse mode?

- Figure S4: It is unclear if the UHSAS volume was corrected or not from the transmission efficiency of the AMS. If not, the scatter plot should certainly provide a similar slope than the ones obtained for Vchem vs Vphys-TC. Why converting UHSAS in mass concentration while all the discussion is made in volume? How is the size distribution after applying transmission efficiency correction?

---

## Author Comment (AC1) · 10 Mar 2021

Please find the response in the attached PDF.

Please also note the supplement to this comment:
https://amt.copernicus.org/preprints/amt-2020-224/amt-2020-224-AC1-
supplement.pdf

---

## Author Response (AR1)

**Response to the comments on the paper "The Importance of Size Ranges in Aerosol Instrument Intercomparisons: A Case Study for the ATom Mission"**

First, we would like to thank both reviewers for their useful comments. Below, we address the reviewer comments and the short comment, in black text and our responses follow in blue text. Changes to the text are shown in **blue, bold text.**

**Responses to Reviewer 1**

**R1.1.** General Comments:

The manuscript by Guo et al. provided a critical evaluation of the size-related factors impacting aerosol intercomparisons and aerosol quantification during NASA Atmospheric Tomography Mission (ATom), with a focus on the Aerosol Mass Spectrometer. This is an important and necessary piece of work, which can fill the absence of significant unknown biases and the appropriateness of the accuracy estimates for AMS total mass/volume for the mostly aged air masses encountered in ATom. Overall, the paper is well-written. One of the major concerns of this study, however, is the reason for choosing certain bin width, which influences particle size distribution and subsequent comparisons. Before its publication, the following comments need to be addressed.

See the response to R1.2 for the bin width question.

**R1.2.** The reason for the choice of bin width (line 337) needs to be elaborated. What is the difference of size distributions in different seasons between using this bin width and using others (narrower or broader width)? The authors need to address such uncertainties in the revised manuscript.

In general, the choice of bin width is based on a balance of resolution and counting statistics, as well as for meaningful visualization of the results. 20 bins/decade is widely adopted in reporting ambient aerosol size distribution data, as is the case for the ATom AMP dataset. Sufficient size resolution is

important to characterize ambient particle composition, which can at times be highly size-dependent (Zhang et al., 2004), and when analyzing phenomena with strong nonlinear transitions vs. particle size, such as cloud condensation nuclei (CCN) activation. Size resolution can also be important for instrumental reasons. For example, using sufficient resolution introduces fewer uncertainties when applying the sloped AMS transmission (i.e., not 0% or 100%) to size distribution measurements. For example, a distribution with 5 bins/decade only has 2 bins in the upper-end AMS transmission size range (482-1175 nm $d_{va}$), while the 10 bins/decade has 4 bins, and the 20 bins/decade has 8 bins in that region. To illustrate that point, in the following figure, we compare $V_{phys,AMS}$ calculated with different size resolutions (10 and 5 vs. 20 bins/decade), for which the AMP size distribution is first converted to the broader bin width by conserving the detected particle numbers, and the AMS inlet transmission is calculated at the diameter of the center of each bin. We find that the broader bin width is associated with more scatter, although the effect is minor in the ATom-1&2 cases since the AMP size distribution is relatively flat in the upper-end AMS transmission size range (see Fig. 2 in the revised manuscript for instance, or Fig. 3 in the AMTD version; the figure number has been changed since we moved Fig. 1 to SI as discussed in the response to comment R2.29). We have added the following text to the manuscript to address this point:

**"Besides, as a sensitivity test, we estimate $V_{phys,AMS}$ based on broader bin widths to test the impact of AMP size resolution. We find that using 10 or 5 bins/decade has minor effects compared to the AMP reported 20 bins/decade (0.4% deviation in slope for 10 bins/decade and ~1.6% for 5 bins/decade), despite the slightly larger scatter as expected from applying AMS transmission to a coarser size distribution."**

It should be noted that the original manuscript already contained a similar analysis for the effect of size resolution on the analysis/representation of the PALMS-AMP data (Fig. S14 in SI of the AMTD

version). This analysis has been further refined in response to the comments from the PALMS team, (see details in our response to comment S1.3).

[Figure]

Fig. I (roman numerals are used for figure numbers in the order in which they appear in this response; this figure is also Fig. S29 in SI): **Comparison of $V_{phys,AMS}$ calculated with different size resolutions for (a) ATom-1 and (b) ATom-2. The AMP size distribution is first converted to the broader bin width, i.e. 10 or 5 bins/decade vs. the reported AMP 20 bins/decade size resolution, and then the AMS inlet transmission is applied.**

**R1.3.** The manuscript focused on size-related intercomparisons, but what are the uncertainties in using the same CE for all particle sizes?

Composition-dependent *CE* calculated based on Middlebrook et al. (2012) was applied to the AMS quantified mass and was near 1 most of the time, a consequence of the high acidity of the ATom observations (refer to Line 165 and Fig. S11 of the AMTD version). Deviation typically originates from external mixtures, such as fresh primary organic aerosol or nitrate plumes, as in some urban

environments. The contribution of POA is very small for the aged and remote aerosols sampled during the ATom (Hodzic et al., 2020). In the ATom context, the two main sources of uncertainty in this area are externally mixed sea salt plumes in the marine boundary layer, and possibly externally mixed OA newly formed in the upper troposphere (Williamson et al., 2019). The latter effect is ruled out due to not observing an altitude-dependent deviation in the volume closure, despite the fact that the Aitken mode contributed up to 50% of the mass in the upper troposphere. In the current manuscript, we have always assumed sea salt to be externally mixed and assumed a $CE$=1 in the moist boundary layer. It is important to note that the uncertainties in size cut, effective density, and $RIE$ of sea salt are likely larger than the uncertainty in $CE$ for sea salt aerosols. Overall, for the intercomparisons presented in this study, we don't observe a deviation correlating with a potential $CE$ effect, thus concluding that such deviation is within the reported 30% uncertainty of $CE$. We plan to evaluate the intercomparison of chemical species in a follow-up paper that might inform this issue further.

**R1.4.** The PALMS+AMP characterize about 54% of Vphys, which is much lower than that in ATom-1 and -2, the SAGA filters, MOUDI, AMS (line 689-691). Please elaborate.

The $V_{phys}$ fraction observed by each instrument is calculated by applying its inlet transmission to the AMP size distribution and averaged over the campaign period. Unlike the other instruments, PALMS-AMP collects less of $V_{phys}$ as altitude increases (Fig. 8) because it doesn't report below 100 nm, and due to the larger fraction of the aerosol population at smaller diameters aloft than at the surface (as shown in Fig. 7 by the AMP measured size distributions). This is more pronounced in the campaign averages since, during ATom, the DC-8 spent more time in the upper troposphere than in the marine boundary layer (Fig. S1 in the revised manuscript, shown below). We made a minor update of this number in the revised manuscript. This number (originally 54%) increased slightly to 56% (see the response to comment S1.10 for the details). Please also note that we also reported the $V_{phys}$ fraction by PALMS-AMP using the 4 bins that assume 100% data coverage over the size range that PALMS reports

(100-5000 nm). That number, 76%, is similar to those of the MOUDI and AMS. The above discussion is in Section 3.6 and we added Fig. S1 to SI. The following changes have been made to the text:

**"The $V_{phys}$ fraction observed by the PALMS-AMP is the lowest because of the opposite trend vs. altitude compared to the other instruments (discussed in the next paragraph), and the larger fraction of the sampling time in the upper troposphere vs. below in the ATom deployments (Fig. S1)."** and **"For the AMP size range, similar increasing fractions of $V_{phys}$ as a function of altitude are observed in all the panels, except for PALMS-AMP, due to the larger fraction of the aerosol population at smaller diameters aloft than at the surface (Fig. 7)** (Williamson et al., 2019)**, since PALMS-AMP does not report below 100 nm $d_p$."**

[Figure]

Fig. II (now Fig. S1 in SI): **Vertical 1-min data coverage during the ATom deployments. DC-8 sampled more often below 800 m and between 9 and 11 km compared to the intermediate altitudes. This sampling distribution is consistent with the generic ATom flight plan (longer**

**sampling in the boundary layer and at the max altitude given air traffic control and fuel/weight considerations) and hence was also consistent throughout the ATom studies.**

**R1.5.** "r" in "r2" should be in italics.

We have revised the text accordingly.

**Responses to Reviewer 2**

**R2.1**. The paper from Guo et al. reports on the importance of the size ranges of both inlets and instruments in aerosol intercomparison with a special focus on airborne measurements during ATom Mission. Such a systematic analysis is essential for better understanding the difference that can be observed when comparing different instruments or inlets to each other, especially during aircraft measurements where altitude can strongly affect the size range of the aerosol inlet and/or instruments. This work is, therefore, suitable for a publication to AMT. However, several issues first must be clarified.

We thank the reviewer for the positive review and the detailed suggestions for improvements, which we address below.

**R2.2.** First of all, the authors must reexamine what is the real focus of the manuscript. Although the title mentions "size range in aerosol instrument intercomparison", abstract and introduction only refer to AMS issues, while only a few words are made on the other inlets and instruments on the conclusion. This does not reflect both manuscript and supplementary information, which are deeply discussing the effect of altitude on the size-range of different inlets and instruments without directly comparing them to AMS results. Consequently, if the manuscript aimed to investigate the effect of the AMS aerodynamic lenses transmission efficiency during intercomparison exercises during ATom Missions, discussions on MOUDI-PM1, URG-PM1 inlets, SAGA and PALM-AMP are not needed on the manuscript since the AMS was not connected to such inlet or no direct compared to them. On the other hand, if the target of the authors was to discuss how the size-range of inlets and individual instruments are affected by altitude during the ATom Missions, then the authors may need to reconsider the discussion on the AMS and the comparison Vchem vs. Vphys or Vphys-TC.

This is a case study for the ATom Mission that focuses on the AMS quantification and the instruments that the AMS is typically compared with. We have expanded the scope of the intercomparisons to also include the widely used MOUDI and URG ground inlets to make the study more useful to the larger AMS and aerosol chemistry measurement communities, since the larger issues discussed are the same regardless of platform (airborne vs. ground). The study highlights the variable inlet transmission between aerosol instruments and the role of altitude/pressure, the latter being important specifically for an aircraft study, but also for higher altitude ground sites. Both the operational size range of an instrument as well as the size range that mostly drives mass/volume is highly variable for most aerosol instruments. So the usual approach of "These are both $PM_1$ instruments, so we can compare them directly" does not work in ATom nor for other studies with highly variable aerosol volume distributions. Hence we are proposing a framework of how to take all these differences into account and exemplifying this on the basis of the ATom mission, including discussing the intercomparisons between AMS and AMP. This paper also serves as the basis for a follow-up study that compares the individual chemical species with the ATom aerosol payloads. To address the reviewer's concern, we have added the following discussion of other instruments in the abstract:

**"The particle size ranges (and their altitude dependence) that are sampled by the AMS and complementary composition instruments (such as Soluble Acidic Gases and Aerosol (SAGA) and Particle Analysis by Laser Mass Spectrometry (PALMS)) are investigated, to inform their use in future studies."**

In the conclusion section:

**"The overlap in the collected particle volumes between the AMS and an aerodynamic $PM_1$ cut, such as the MOUDI 1 μm stage impactor (dry condition; AMS vs. MOUDI 85 ± 10%) or SAGA**

**MC (ambient condition; AMS vs. SAGA MC 97 ± 14%), suggest a direct comparison of these bulk aerosol properties is generally meaningful."**

**R2.3.** According to the title, I would expect more direct intercomparison between the different results providing by the instruments, similarly than for the AMS (Vchem vs Vphys-TC) rather than simply comparing numbers or transmission profiles. For example, why the authors did not compare their AMS results with SAGA, and PALMS-AMP? how the change in transmission efficiency of each instrument affect the comparison?

A meaningful chemical comparison needs to be based on the understanding of inlet transmissions (i.e., what fractions of aerosols we are comparing, and how this changes with altitude and time). As suggested by the reviewer, we are currently working on a follow-up paper discussing the intercomparison of the chemical measurements of AMS, SAGA, and PALMS-AMP. Hence, the discussion about the inlet transmission in this study serves as an important basis for comparing AMS to PALMS-AMP since if a volume closure between AMS and AMP is not observed after accounting for the AMS inlet transmission, a further intercomparison of the PALMS-AMP derived mass products based on AMP would not be meaningful.

**R2.4.** Only very spare information is done on the different methods used to estimate the change of the size-range of the instrument with the altitude. It would be great to have, for example, a dedicated section on it detailing the different hypotheses and models used for the calculation.

The method for calculating $d_{50}$ for changing pressure and temperature are discussed in the Table S1 caption. We did not make it clear that other sizes (than $d_{50}$) were calculated following the same procedure. Therefore, we have added a paragraph to the revised SI to explain this point:

"For AMS, the cut sizes in $d_{va}$ are native and the other two sizes ($d_{ta}$, $d_p$) are calculated with Eqs. 1-2 in the main text. For URG, MOUDI, and SAGA MC, the cut sizes in $d_{ta}$ are native, since the size selection is normally conducted in the transition regime. Specifically, given the native cutoff size in $d_{ta}$, the cut sizes in $d_p$ for MOUDI are calculated using Eq. 5.28 in Hinds (2012):

$d_{50} = \sqrt{\dfrac{9\eta D_j (Stk_{50})}{\rho_p U C_c}}$. For circular jets such as MOUDI, 50% collection efficiency corresponds to Stokes Number, $Stk_{50}$, of 0.24. $\eta$ is air viscosity, $D_j$ is the nozzle size (0.78 mm) (Marple et al., 2014), $U$ is air velocity (a nominal volumetric flow of 30 L m$^{-3}$ gives 26.16 m s$^{-1}$ with 40 nozzles at the size of 0.78 mm). The equation is also used to estimate the $d_{50}$ for SAGA MC by dividing the formulas between the two conditions, and the base case gives $d_{ta,sea,50}$ of 1 μm (van Donkelaar et al., 2008) (discussed below at Sect. 10). Note that this method is not limited to calculating $d_{50}$ but also other sizes so that the inlet transmission profile can be estimated for different pressure, temperature, or aerosol density, given an initial profile under known conditions."

**R2.5.** Focusing on the AMS, the authors did not convince me of the importance of transmission efficiency. Although the focus of the manuscript (based on abstract and conclusion) is to demonstrate the importance of the transmission efficiency of the aerodynamic lenses of the AMS during intercomparison, it is first mandatory to ensure, before any comparison that all the reference instruments are properly calibrated. Here, the calibration and quality assurance of the different optical size spectrometers are not mentioned. How do the 3 instruments compare on their overlapping size-range? The manuscript will certainly gain consistency if the authors can also discuss these points as well as their uncertainties rather than only citing literature (lines 301-307) similarly than for the AMS. In such an exercise, not only the AMS had to be carefully calibrated.

It is an excellent point brought up by the reviewer that, to ensure a meaningful comparison, both instruments being compared must be carefully calibrated and operated. Often, operational or analytical

errors could be misinterpreted as instrumental biases. The calibration of the AMP instruments for ATom specifically was discussed in the citations provided (Brock et al., 2019; Kupc et al., 2018), and the latter reference discussed the overlap of the instruments in detail. We see no reason to repeat that information in this manuscript. We added the following text to the AMP method part to explain the consistency between UHSAS and LAS in the overlapped size range:

**"AMP performed well and consistency was found in the overlapping size range during ATom. For instance, Brock et al.** (2019) **found agreement within 1% for particle number and 9% for integrated volume for the overlap between the UHSAS and LAS during ATom-1. Although the NMASS barely overlapped with the UHSAS in size coverage, the two size distributions appear to agree well with each other as shown in Fig. 6 in Brock et al.** (2019)**."**

**R2.6.** Another critical aspect is the absence of discussion when comparing correlations with and without applying the size-dependent correction method. Figure S4 shows a very good correlation between the instruments without applying it. The discussion on the effect or not of the size-dependent correction should be supported by quantitative numbers as well to demonstrate that transmission efficiency of the AMS plays an important role (which was up to now never considered on the literature) and properly quantify how much Vchem can be affected by the transmission efficiency of the AMS lenses. The reason is certainly on the definition of the Vphys itself. Why Vphys has a volume concentration ranging up to 4.8 µm? This is quite surprising since there is a long discussion on the change of the size-range of the AMS between the missions (section 3.1 and 3.2). Moreover, the large fraction of super-micrometer particles on the volume size distribution is controlling most of the total volume mass concentration on Vphys. Why the authors did not cut Vphys to 1 µm for all missions? Or more simply using Vphys-0.75 or Vphys-0.95, corresponding to their own estimated AMS upper-size-cutting (PM0.75 for ATom 2 and PM0.95 for ATom2, for example). It would also be possible to consider a volume size distribution up to 800 nm (equivalent diameter similarly to most of the SMPS system). Applying such kind of approach

provides i-/ a comparable size-range between the 2 instruments and ii-/ the possibility to extrapolate the results to most of the AMS work that was done up to now, which are mostly using SMPS for comparison. Only then, the authors can properly discuss the importance of considering or not the aerodynamic lens transmission efficiency on the comparison with collocated measurements, and how much it can influence the correlation. It would be also interesting to do a sensitivity test based on the influence of the coarse mode concentration (low and high coarse mode periods). For this purpose, Table S2 is certainly one of the most important for the comparison between the different instruments. However, I don't fully understand why only Vphys-TC is used and not a systematic Vchem and Vphys-TC on a similar size range to quantify the effect of the AMS transmission efficiency correction on different systems.

The important role of AMS transmission efficiency has been discussed multiple times previously in the literature (Hu et al., 2017, 2020; Knote et al., 2011; Nault et al., 2018; Poulain et al., 2020; Saide et al., 2020; Schroder et al., 2018), although not as extensively as in this study. The effect of applying the AMS inlet transmission was clearly illustrated as the time series in Fig. 3c. We have added the comparison between $V_{chem}$ and $V_{phys}$ in Fig. 3a&b (and the inset of panel c), as gray markers, to show the effect of applying the AMS transmission to $V_{phys}$. The following text is added,

**"The effect of applying the AMS transmission to $V_{phys}$ is also shown in Figs. 3a&b as the gray markers on the campaign level. Clearly, at times the effect is major, and at other times minimal, depending on the ambient size distribution."**

[Figure]

Fig. III (now Fig. 3 in the main text): **Comparison between $V_{chem}$ and $V_{phys,AMS}$ for (a) ATom-1 and (b) ATom-2, data points colored by altitude, and averaged to 5 min resolution. $V_{chem}$ is also compared to $V_{phys}$, as the gray markers, to show the effect of not applying the AMS inlet transmission. (c) A time series of these volumes for a research flight in ATom-2, with an inset showing the scatter plot only for this flight (at 1 min time scale, as also shown for the time series). Note that $V_{chem}$ includes the AMS quantified sea salt. Two correlation coefficients ($r^2$) are listed: one at linear scale (commonly used) and the other at logarithmic scale, which emphasizes the scatter at low concentrations.**

As the reviewer pointed out, we used the whole size range of $V_{phys}$ up to 4.8 μm but did not cut $V_{phys}$ at a certain size, such as 1 μm, although we considered this option. The reasons include: (1) Choosing an intermediate cutoff size could be arbitrary and not supported by clear scientific reasons. The size range of SMPS varies depending on the model or customization (e.g., the SMPS size range was 0.012-0.67 μm in the MILAGRO study (DeCarlo et al., 2008), 0.013-1.1 μm in SOAS (Nguyen et al., 2014), 0.01025-1.094 μm in ACTRIS (Crenn et al., 2015), and 0.009-0.85 μm in Bougiatioti et al. (2014)). (2) Aerosol density plays an important role in shifting the equivalent $d_{ta,sea,50}$ given an AMS transmission curve (as shown in Fig. S26). The same ATom-2 inlet transmission has a $d_{ta,sea,50}$ of 599 nm at 1.7 g cm$^{-3}$, while this number increases to 789 nm for fresh oily OA at 0.9 g cm$^{-3}$. Thus, it is necessary to apply the inlet transmission for a proper intercomparison. (3) Using the whole size range makes the intercomparison to other instruments consistent.

To account for the fact that a large fraction of coarse mode particles does not pass through the AMS lens, we show the results from MOUDI 1 μm stage impactor and URG PM$_1$ cyclone and compare them to AMS (Fig. 6). We believe using the transmission characteristics of those real instruments is more meaningful than an arbitrary "vertical" cut that would not correspond to the way the SMPS is operated in most studies.

We have added the following text to Sect 3.3 to address this point:

**"When AMS transmission is not characterized, an alternative for volume intercomparison is to truncate $V_{phys}$ at a certain size (e.g., 1 μm). In this case, the intercomparison is not ideal (shown as Fig. S31 with slopes of 0.74 and 0.65 for ATom-1 and -2, respectively, with more scatter for ATom-1), highlighting the importance of calibrating and applying the inlet transmission."**

[Figure]

Fig. IV (now Fig. S31 in SI): **Comparison between $V_{chem}$ and $V_{phys,<1\mu m}$ for (a) ATom-1 and (b) ATom-2. Compared to Fig. 3 (main text), the plotted $V_{phys,<1\mu m}$ is truncated at 1 µm ($d_p$) of $V_{phys}$ to simulate a simpler comparison of volume concentrations without applying AMS transmission profile. Data points are colored by altitude and averaged to 5 min resolution.**

**R2.7.** Does it make sense to apply the transmission curve on Vphys? This approach makes Vchem and Vphys dependent variables. Furthermore, assuming the size spectrometers have perfect transmission efficiency and particle detection up to 4.8 µm, it is not completely surprising that after applying the AMS transmission efficiency correction, a perfect correlation is obtained. Finally, it does not say that the two instruments better correlate than without applying the correction. Instead, it would be better to correct the AMS mass or volume concentrations by the corresponding transmission efficiency. This means first ensure that ePTOF and MS mass concentrations agree, then correct the ePTOF mass concentration from transmission efficiency to get the final AMS mass concentration. Then, it would be possible to compare to parallel measurements of the same size cutting. However, I have to recognize that I am not sure whether this is feasible on this dataset or not.

$V_{chem}$ and $V_{phys}$ are mostly independent of each other, since $V_{chem}$ is derived from mass-based measurements (AMS and BC) while $V_{phys}$ is derived from number-based measurements (AMP). The best estimate of the transmission curve (based on transmission calibrations) is applied to $V_{phys}$ mathematically, while the AMS instrument independently "applies" the real transmission curve to the particle population itself. Another variable used when applying the AMS inlet transmission to $V_{phys}$ is the density of the aerosol, which is needed since all the inlet transmission depends on it. While density can be estimated by other methods, the only available measurement in ATom is from the AMS (w/ SP2 BC). Since the density is an intensive property, which is a complex function of fractional composition and the oxidation state of OA, and since this density is not used in any way to calculate the AMS mass concentration, we strongly disagree that the comparison involves "dependent variables."

To further document and explain the AMP size coverage, we have added the following text to the paper:

**"AMP gives nearly unity detection efficiency of the ~5 nm to ~4 μm aerosols at sea level: (1) The NMASS had nearly unity detection efficiency from ~5 nm to 100 nm but only reported up to 60 nm; (2) the UHSAS had > 90% counting efficiency from 63 to 1000 nm; (3) the LAS had high detection efficiency between 120 nm and 10 μm, however, the max size was limited to < 4.8 μm by the aircraft inlet (**Brock et al., 2019**)."**

Its inlet has the largest cutoff size among the aerosol instruments. Therefore, the AMP data can be used to represent the in-situ particle (dry) size distributions for that size range and allows us to evaluate how much each instrument inlet observes by applying a corresponding inlet transmission.

Finding agreement between AMS and AMP after applying the AMS transmission is not guaranteed. The comparison can go wrong for many reasons, such as wrong *CE, RIE* applied to the AMS data, other

errors in AMS data processing, or any errors or biases from AMP. Therefore, the agreement supports that the two independent instruments provide accurate quantification themselves (at least for the overlapping size range) and that the AMS transmission is properly calibrated (and also, the estimated or assumed densities of aerosol species are reasonable).

As the reviewer correctly implies, any ePToF data from the AMS should be affected similarly by the AMS inlet transmission as the MS data is. Correcting the ePToF size distribution by the AMS inlet transmission is, however, practically difficult or impossible for three reasons. (1) It is almost impossible to reconstruct ambient particle size distribution when AMS has negligible transmission and this data inversion process will greatly magnify the uncertainties or biases in the ePToF measurements. (2) For instance, any PToF measurement of OOA and sulfate is typically affected by some vaporization delays at the high end of the distribution that smear it out to larger sizes and is therefore unsuitable to be used to estimate transmission correctly (although it is certainly useful for confirmation). (3) concentrations were very low during most of ATom and ePToF typically has a low signal-to-noise ratio (S/N) except for longer averages. But even if the ePToF in ATom had the adequate S/N to do what the reviewer proposes (which it does not), as described we see no benefit in using the reviewer's method over the one presented in this paper.

**R2.8.** Although, the authors carefully investigate the transmission efficiency of their instrument; key information on the calibration protocol itself is not mentioned. For example, how the particles were generated and selected (especially for mono-dispersed super-μm ammonium nitrate particles)? Which instrument was used as a reference in parallel to the AMS? The authors used 400 nm (mobility) ammonium nitrate particles to perform their IE calibrations, however, looking at Figure 2, this diameter seems to be already out of the 100 % transmission efficiency of their aerodynamic lenses having rather a transmission ranging from 80-95%. How did the authors deal with that? Does it mean that their RIE (and the resulting AMS mass concentration) is underestimated?

We have added additional discussion of the calibration procedures to the text to document these topics:

**"After every research flight, $IE_{NO3}$ was calibrated by atomizing pure $NH_4NO_3$ solutions and selecting dry (desiccated with a Nafion dryer) 400 nm (mobility diameter, $d_m$; equivalent to $d_{va}$ = 550 nm)** (DeCarlo et al., 2004) **particles with a differential mobility analyzer (DMA, TSI model 3081, St. Paul, MN, USA) into the AMS."**

**"Besides these single-size (at the edge of the $E_L$~1 range) post-flight calibrations, the upper end of the AMS transmission curve was characterized on the aircraft during ATom-2 by measuring multiple sizes of monodisperse ammonium nitrate ($d_m$ range 350-850 nm) by comparing the mass measured by AMS to that by CPC (i.e., CPC counts × single-particle volume). Multiply charged ammonium nitrate aerosols were removed by the impactor upstream of the DMA, which was confirmed by the AMS size-resolved measurements."**

The $IE_{NO3}$ was based on the single-particle-based transmission calibration protocols (as documented e.g. in the presentations in the AMS Users Meeting and Data Analysis Clinic (DeCarlo (2009) on Brute Force Single Particle (BFSP) and Nault (2016) on event trigger (ET)). Please note that we are also working on an AMS operation and calibration technical paper (led by B. Nault), aiming to provide detailed field operation guidance, and further documenting these procedures. The AMS had ~85% inlet transmission for 400 nm ($d_m$) ammonium nitrate particles in ATom-1&2 but that doesn't affect $IE$ single-particle-based calibrations. $IE$ is calibrated by the ammonium nitrate particles that are transported to the vaporizer and individually detected. So as long as the size of the particles is indeed correct (which we confirm by both PToF and PSL calibrations of our calibration SMPSs), a valid $IE$ can be derived, since it is completely independent of the total amount of particles making it through the lens.

**R2.9.** Last but not least, the reported discussion regarding PALMS and PALMS-AMP analysis must be clarified and solved on the revised version of the manuscript.

We address the concerns regarding the PALMS-AMP analysis in detail below.

**R2.10.** Minor Comments: "The abstract is rather unspecific and gave poor information on the "critical evaluation of the size-related factors", the "good agreement" when comparing AMS with "size spectrometers" and "standard PM1 volume". Without any indication of the transmission efficiency, the size range of the reference used, and results before and after applying the correction, how can we conclude from an "absence of significant unknown biases"? More details on the "upper end of the AMS size-dependent transmission" as well as the upper size cutting of the size-spectrometer must be provided here.

We have revised the abstract to address these points. We revised the "critical evaluation of the size-related factors" to "**critical evaluation of inlet transmission**" to be more specific. Also, we added "**(regression slope = 0.949 and 1.083 for ATom-1 and -2, respectively; SD = 0.003)**" after the "good agreement" and "**(referred to a URG standard cut 1µm cyclone operated at its nominal efficiency)**" after the "standard PM$_1$ volume". The statement of not finding significant unknown biases of the AMS quantification is supported by the volume closure between AMS and AMP (after applying the AMS transmission). The details of the AMS transmission and the upper cutting size of AMP are too technical for the abstract, so we did not add them.

**R2.11.** Why results from the other instruments (SAGA, PALM) and inlets (MODI-PM1 and URG-PM1) are not mentioned in the abstract?

Those results are important, but the abstract is limited in length, and those items seem too technical to be discussed in the abstract in detail. The last sentence in the abstract already pointed out that other instruments were also discussed in this study. We have added more detail on these instruments to the abstract, as discussed in response to comments R2.2 and R2.10.

**R2.12.** line 28: What is a good agreement? Please provide number.

See response to comment R2.10.

**R2.13.** line 29: Please precise from which size-spectrometer (and its size-range) were used.

We have changed the text to:

**"The volume determined from physical sizing instruments (Aerosol Microphysical Properties, AMP, from 2.7 nm to 4.8 µm optical diameter) is compared in detail with that derived from the chemical measurements of the AMS and the Single Particle Soot Photometer (SP2)."**

**R2.14.** line 31: What is a "standard PM1 volume"?

See response to comment R2.10.

**R2.15.** Introduction: Over the last two decades, there is a lot of works reporting comparison between AMS and collocated instruments either for individual chemical species and/or total mass/volume. Most of the time the correlations are within the different instrumental uncertainties. Therefore, the authors should provide more clear justifications for their motivations and the need for such correction on the introduction.

Yes, there have been a lot of intercomparisons involving AMS in the past, but most do not go further than doing the scatter plots or time series comparisons with other collocated instruments, because intercomparison is not the main focus of those studies, and often because the quality of the AMS dataset and those of collocated instruments is not sufficient for very detailed analyses. Also, some mixed results have been reported, many with agreement within the reported uncertainties, and some not. This study aims to examine these intercomparisons rigorously using a suite of high-quality datasets, and with a focus on the different size coverage of the different instruments. An important aspect is the AMS lens transmission, which is often not calibrated for, and can lead to confusing apparent disagreements in some studies.

**R2.16.** line 53: ACTRIS and not ACTRiS

Thanks for the correction.

**R2.17.** line 65: Could you clarify the sentence "Volume comparisons probe the ability of the AMS to quantify total mass and predict aerosol density based on fractional composition accurately, and hence it the most germane comparison for total quantification"

Comparing volume (unlike say, extinction measurements that depend both on size and optical properties, or chemical comparisons with instruments that often do not measure exactly the same chemical species as the AMS) is the most direct way to evaluate the accuracy of the AMS mass (and to some extent, the estimated density from AMS measurements), especially since particle sizers, in general, have the least amount of assumptions in their data products. We have revised the text to clarify these points:

**"Volume comparison probes the ability of the AMS to quantify total aerosol mass and predict aerosol density (based on fractional composition) accurately, and hence is the most direct method to evaluate the AMS overall quantification (unlike e.g. comparing total mass to extinction that depends on mass extinction efficiency)."**

**R2.18.** line 73-75: Does it make sense to repeat this information. This was already mentioned in the introduction.

We have deleted the redundant statement.

**R2.19.** line 112: Details on the modifications made on the AMS compared to a "standard" one would be appreciated

Some modification details are already discussed in Sect. 2.4 "AMS operation during ATom", such as the extensive discussion on the AMS inlet configuration and performance (Sect. 4 in the Supp.) and the implementation of a cryopump to reduce noise from instrument background (which we hope to discuss in more detail in a future manuscript). Many of the other modifications are discussed in previous papers, as discussed in this text from the methods section of the AMTD version: "The aircraft operation of the CU AMS has been discussed previously (Cubison et al., 2011; DeCarlo et al., 2006, 2008, 2010; Dunlea et al., 2009; Kimmel et al., 2011; Schroder et al., 2018). The specific operational procedures used during ATom have been discussed in Nault et al. (2018) and Hodzic et al. (2020). Important operation details of AMS that are relevant to this study are described below." Therefore we have not expanded the manuscript further on this point.

**R2.20.** line 112 – 284: I think a detailed description of the AMS is not needed. AMS is not a new instrument and it was already described in the cited papers. Moreover, I don't follow how section 2.3 is

splitting. For example, the authors first discuss the factor influencing the collection efficiency (line 125-145) and then with the impact of the RH and the CDCE (line 194-199). A similar comment can be made for the IE and RIE. The current splitting leads to unnecessary repetitions. For example line 143-144: "CE is estimated to contribute substantially to the overall uncertainty of the AMS concentration measurements (Bahreini et al., 200), line 169-170 "For the AMS reported mass concentration, uncertainties (i.e. accuracies) in CE (30%)", and line 211 "[. . .] are similar to those from Bahreini et al. (2009), due to the dominate uncertainties contribution from CE (30%)". Similarly, the uncertainties of the RIE is discussed on line 157-172 and again online 204-211. A single AMS section will improve it.

Based on the reviewer's comments, we have merged or deleted the redundant statements in the two sections. Sect. 2.3 provides a general introduction of the AMS quantification and Sect. 2.4 provides more technical details during the ATom deployments. Although the general aspects of AMS have been discussed very well in the literature, it is useful to re-describe the same topic briefly with a focus on the aspects that matter the most in this particular study. For instance, $E_L$, the lens transmission and part of *CE*, is often not characterized specifically in many studies. The discussion on $RIE_{OA}$ illustrates why using a constant value of 1.4 is reasonable for ATom because of the fairly aged aerosols sampled in this study. Merging the two sections will deemphasize such points since Sect. 2.4 is already long.

**R2.21.** line 269: density calculation (eq. 5): why not directly include sea-salt since i-/ you quantified it on your AMS measurements (line 256) and ii-/ you include it anyway later (line 382-386). A similar comment can be made for rBC. Then you can discuss once how total mass concentration and density were estimated. Here again, it will improve the clarity of the manuscript.

We have made a new section (as Sect. 2.6) discussing the aerosol density estimation, including sea salt and rBC in the equation.

**"For the purpose of instrument comparisons, we estimate the aerosol volume based on the chemical instruments ($V_{chem}$). $V_{chem}$ is determined from the AMS non-refractory mass concentrations plus the refractory species sea salt and rBC by assuming volume additivity, with an average particle density estimated as** (DeCarlo et al., 2004; Salcedo et al., 2006)

$$\rho_m = \frac{OA+SO_4+pNO_3+NH_4+Cl+Seasalt+rBC}{\frac{OA}{\rho_{OA}}+\frac{SO_4+pNO_3+NH_4}{1.75}+\frac{Cl}{1.52}+\frac{Seasalt}{1.45}+\frac{rBC}{1.77}} \tag{1}$$

..."

**R2.22.** line 272: Density of OA was calculated based on the Elemental Analysis of the organic mass spectra ranging from 1.51 – 1.59 g cm-3 (line 280) but the default OA density is set to 1.7 g cm-3. Does the default value make sense? Could it be that the value of 1.7 is artificially high due to low concentration and overall strong contribution of CO2+ to the total mass spectra?

The 1.7 g cm$^{-3}$ OA density is applied when OA mass concentration is below the DL and the parameterization method no longer works. This density is found around the DL and it is slightly higher than the campaign average because these low OA masses tend to be more oxidized. We have added the following figure to SI to document this point:

[Figure]

Fig. V (Fig. S20 in SI): **The ATom-1&2 OA atomic ratios (H/C, O/C) and density vs. concentration. The shaded regions represent the precision as a standard error. This figure shows an increasing trend of $\rho_{OA}$ at lower OA mass concentrations. As Fig. S21 shows high $\rho_{OA}$, such as 2.0-2.3 g cm$^{-3}$, is only predicted for a small percentage of the data points. Also shown is a similar trend of increasing density vs. mass loading, with similar values for the same OA concentrations, previously reported in laboratory SOA studies** (Shilling et al., 2009)**.**

**R2.23.** line 301: Vphys is a combination of 3 optical particle size-spectrometer ranging from 2.7 nm to 4.8 µm. How these 3 instruments are comparing together on their overlapping size range?

Please see our response to comment R2.5.

**R2.24.** line 358: Do you need to include the SAGA on the manuscript since it is no use for the comparison with the AMS?

Yes. This manuscript is part of a full evaluation of the aerosol payload in ATom, and the comparison between SAGA and AMS will be discussed in a follow-up paper.

**R2.25.** line 441: Is this section necessary, since none of the instruments used for the discussion were connected to such inlets?

The manuscript makes the case that "PM$_1$" to describe the AMS size range is in many cases not precise enough and this matters for intercomparisons, both for airborne measurements and on the ground. Hence this section is very relevant, especially to the majority of AMS users that don't participate in airborne campaigns.

**R2.26.** line 515: the two following sentences are unclear and need to be rephrased "AMS observes a fraction of Vphys (Figs. 1 & 3). AMS transmission vs. dp was calculated based on the calibrated transmission vs. dva (Eq. 1) and the AMS estimated time-dependent ρm (Eq. 5), and applied to the AMP size distributions used to derive Vphys."

We have revised this text to read:

**"AMS observes a fraction of the full AMP size distributions, as shown in Fig. 2. To properly characterize the part of $V_{phys}$ observed by the AMS, we first convert the calibrated $d_{va}$-based AMS transmission to its $d_p$-based form (using Eq. 1) with the time-resolved $\rho_m$ estimated from the AMS (Eq. 5). This volume is referred to as $V_{phys,AMS}$ (the AMS-transmission-corrected $V_{phys}$)."**

**R2.27.** line 520: How is the correlation without corrected Vphys by AMS transmission?

We have added the comparison between $V_{phys}$ and $V_{chem}$ to Fig. 3 in the revised manuscript, as discussed in the response to comment R2.6.

**R2.28.** line 523: "slightly worse"? Could we say that this correlation is slightly worse according to all instrumental uncertainties? Both are within 10%. Instrumental uncertainties on the comparison should be discussed. Which type of regression was used (classical linear fit or least orthogonal distance fit)?

The regression slope of ATom-2 is further from 1 than ATom-1 and the $r^2$ is 0.93 vs. 0.95, so we summarized that result as "slightly worse". As the reviewer pointed out, the slope is well within the combined uncertainties of the two instruments. We discussed that in detail in Line 557. We have reworded the sentence to,

**"The fitting slope being slightly further from 1 (1.083 ± 0.003) in ATom-2 ($r^2$ of 0.93) may be due to the larger contribution of sea salt in ATom-2 in the boundary layer** (Hodzic et al., 2020) **and hence the larger uncertainty arising when applying the AMS size cut. Nevertheless, the slopes for ATom-1&2 are well within the combined instrumental uncertainties (discussed in this section)."**

The slope from orthogonal distance regression (ODR) has been used throughout the study for comparing two variables with uncertainties, and the $r^2$ is from the classic linear regression since ODR doesn't give $r^2$. We have added the following text to the methods section to clarify this point:

**"When comparing $V_{chem}$ to $V_{phys}$, orthogonal distance regression (ODR) is always used to derive the linear fitting slope (for comparing two variables with uncertainties), such as in Figs. 4-6."**

**R2.29.** Figure 1: Please use a log-scale for dN/dlogd to better see the particle number > 100 nm.

Thanks, we have updated the figure as suggested. The updated figure is included in the SI as Fig. S23 and shown below:

[Figure]

Fig. VI (Fig. S23 in SI)

**R2.28.** Figure 4: Can the strong increase of Vphys compare to Vchem at the lowest altitude simply be associated with marine sea-salt coarse mode?

We showed the other version of Fig. 3 color-scaled by sea salt mass concentrations in the AMTD version Fig. S14, which shows that sea salt can be important at times to increase $V_{chem}$ but not in all

cases. For example, several of the points with highest $V_{chem}$ in ATom-1 have very limited impacts from sea salt despite being at low altitude.

**R2.29.** Figure S4: It is unclear if the UHSAS volume was corrected or not from the transmission efficiency of the AMS. If not, the scatter plot should certainly provide a similar slope than the ones obtained for Vchem vs Vphys-TC. Why converting UHSAS in mass concentration while all the discussion is made in volume? How is the size distribution after applying transmission efficiency correction?

The UHSAS volume is not corrected by the AMS inlet transmission since it was not fully characterized in the field during DC3. Here we plot the ATom-1&2 transmission just for reference. The key point for this figure is that the comparison between AMS and UHSAS is not affected by the AMS HIMIL inlet, so it doesn't impose any further particle loss on the basis of the AMS lens and the downstream plumbing of HIMIL. The mass-based comparison is used here since the AMS measured submicron composition is shown, which is natively in a mass basis. And mass can be easily compared to extinction (included in the plot) to obtain mass extinction efficiency. In the end, the mass-based and volume-based intercomparison between AMS and UHSAS are equivalent. To make it clear, we have changed the figure caption to:

**"Top: Comparison of AMS speciated mass with UHSAS volume (not corrected with the AMS transmission) and PM$_1$ 550 nm scattering (both operated by the NASA Langley Group) for a period during Research Flight 4 of the NSF/NASA DC3 Mission** (Barth et al., 2015)**, onboard the NASA DC-8, with the AMS sampling line being switched between the AMS HIMIL inlet and the LARGE inlet (the same inlet that the AMP Group used during ATom). Bottom: Average UHSAS volume distributions for the five periods shown. While comparisons like these were performed repeatedly during ATom with similar results, this one was chosen for illustration, due to the much**

**higher concentrations and availability of concurrent optical measurements. The ATom-1&2 inlet transmission is also shown as an illustration since it was not fully characterized in the field during DC3."**

**Additional changes**

The lower end AMS transmission for small particles was adapted from literature for the last submission, as we were not able to access our laboratory to complete the experiments during the COVID-19 lockdown. However, we were able to perform this calibration successfully in the summer of 2020 once we regained access to our lab using a newly constructed evaporation-condensation particle generator for generating monodisperse particles in the range of 20-250 nm (Kim et al., *in prep*). The calibrated curve is similar, although slightly better (i.e., transmission decreases at slightly smaller diameters) than the curve from the literature references (Knote et al., 2011; Zhang et al., 2004). This curve is now shown in Fig. 1 (reproduced below), and all the related results and figures have been updated. However, this update causes minimal change to the graphs and the conclusions since the change is small and only a small amount of mass was observed in the 30-100 nm $d_{va}$ size range.

[Figure]

**Fig. VII (Fig. 1): Results of AMS transmission calibrations vs. vacuum aerodynamic diameter ($d_{va}$) for ATom-1&2. The small particle transmission was calibrated with oleic acid post-campaign (left) and the large particle transmission was calibrated with $NH_4NO_3$ particles in the field. On the right side, the green markers are multi-size field calibrations, and the grey cross markers are single-size (at $d_m$ = 400 nm, equivalent to 550 nm $d_{va}$) field calibrations after every research flight. The insets show the frequency distributions of measured transmissions (right, top) and observed, normalized size distributions (left, bottom) of these single-size calibrations. A fit shows 100% transmission at 483 nm (1σ uncertainty of the fit: 445-525 nm) and 0% transmission at 1175 nm (1σ: 1112-1241 nm). When forcing 0% transmission at 1175 nm (confirmed by $(NH_4)_2SO_4$ calibrations), the fit to all data gives 100% transmission at 482 nm (1σ: 479-485 nm, not shown), consistent with the 483 nm inferred based only on the ATom-2 multi-size field calibrations.**

We added the following text to the AMP method section to explain the differences in the combined ATom size distribution data sets compared to Brock et al. (2021).

"**Other than the AMP that was operated in-cabin of DC-8 and provided dry particle size distributions, the 2$^{nd}$ generation Cloud, Aerosol, and Precipitation Spectrometer (CAPS) was installed underwing to monitor the aerosol and cloud droplet size distributions at near-ambient conditions** (Spanu et al., 2020)**. Since the CAPS has limited size resolution and coverage in the submicron size range that matters the most for the analysis presented in this work, CAPS data in this manuscript is only used to screen for in-cloud sampling. Brock et al.** (2021) **combine the data from AMP and CAPS to derive a size distribution product that covers a wider size range, in which the CAPS data is used above 1.01 μm (and up to 50 μm) and the LAS data is used between 0.50 and 1.01 μm. In this study, the LAS data is used between 0.50 and 4.8 μm** (Brock et al., 2019)**.**"

Lines 331-357 (in the PALMS method section) in the AMTD version have been moved to SI to accommodate the added figures and text, to address the short comments by Murphy et al., since these discussions may be too technical for most readers. Therefore, we added the following text to the PALMS methods part,

"**Other than the allocated four size bins that assume 100% data coverage** (Froyd et al., 2019)**, we also characterize the operational size coverage of PALMS based on the reported size resolution of the AMP particle size distributions (i.e., at higher size resolution) for ease of comparison with other instruments. The details can be found in Sect. 8 of SI.**"

**Responses to Murphy et al.**

**S1.1.** The authors of this comment are the principal investigators for the PALMS single particle mass spectrometer and the aerosol microphysical properties (AMP) size distribution instruments that were operated on the Atmospheric Tomography Mission (ATom). The PALMS authors are the developers of the instrument and associated data analysis methodologies, and have applied the instrument to study the atmospheric aerosol for more than 2 decades. We write out of concern that this manuscript (Guo et al., 2020; hereafter G20) has serious flaws in the way it represents the PALMS instrument capabilities and the PALMS+AMP data products submitted to the ATom database. Others who use data from ATom and other missions could misunderstand how to utilize PALMS+AMP products in the mission archive. Moreover, G20 introduces into the literature a new "chemical information" metric that has unphysical behavior and lacks mathematical rigor.

We respectfully disagree with these statements. As will be shown below, our analyses are now completely consistent with those of the PALMS team, thanks to a minor update based on the data posted in Murphy et al. (2020), which we did not have access to before. In our opinion, these analyses are crucial to understanding the strengths and limitations of the ATom aerosol dataset. In addition, we want to stress that no "chemical information" metric is introduced at any point in the manuscript. This appears to be a misunderstanding but clearly is used in the comment by Murphy et al. as a rebuttal, when it is really a logical "straw man" (https://en.wikipedia.org/wiki/Straw_man). Adapting the Wikipedia definition to this situation: "The typical straw man argument creates the illusion of having completely refuted a proposition (here, the actual analysis of size coverage in this paper) through the covert replacement of it with a different proposition (here, the "chemical information metric") and the subsequent refutation of that false argument ("knock down a straw man") instead of the actual proposition.

**S1.2.** This comment will show that that G20: - Analyzes the PALMS data with arbitrary size bins that are too narrow. The combination of a very high size resolution and prescribing zero values to any bins without PALMS particle observations biases the results. - Includes a conceptually flawed "chemical information" metric that leads incorrect conclusions. - Improperly compares instrument performance by using different metrics for different instruments. - Is quantitatively incorrect in several ways: underestimating by large factors the number of particles actually sampled by PALMS, and producing concentrations that are biased low and scale unphysically with sampling time….Because of the use of PALMS data, K. Froyd and D. Murphy were originally invited to be coauthors on this manuscript but we were removed as authors when agreement could not be reached.

This response will show that:

- The chosen 20 bins/decade size resolution is not arbitrary; rather, we use the reported AMP size resolution to characterize the inlet transmission of all instruments, including the PALMS, to make the most meaningful comparisons possible. Results are similar for other relevant resolutions. See response to comment S1.3 below for details.

- We did not define a "chemical information" metric. See response to comments S1.1 and S1.4.

- The comparison across different instruments is proper and consistent. We stated that all of the observable size ranges by all the different instruments do not consider the details of the chemical detection of any instrument, such as what species is reported or the associated DL for a given species. See response to comment S1.7 for details.

- We strongly disagree that our estimation under-reports PALMS counts by large factors. It is now highly consistent with the PALMS team's data. See response to comment S1.10 for details.

- We reached out repeatedly to the PALMS team to explain that there were no errors in the manuscript, and to carefully consider their concerns, both before and after submission. While we regret that the PALMS team decided to remove themselves from the coauthor list, that was their decision, not ours.

**S1.3.** Arbitrary and narrow size bins. G20 show results obtained by combining PALMS and AMP data (e.g., Fig 8, 9, S28, S29, S30). However, they explicitly do not follow the data analysis procedures used by the PALMS+AMP team for data submitted to the mission archive. Instead, the manuscript "provides an alternative illustration of PALMS size coverage" (line 339). Instead of the 4 size bins used by the PALMS+AMP team to derive absolute concentrations, G20 use 20 bins per decade of diameter, or 35 bins across the nominal PALMS size range. The choice of bin width is critical because it significantly affects their analysis of data coverage and derived concentrations. The G20 choice has no physical basis. Presumably, it arises because the AMP team uses 20 bins per decade as a convenient bin size to report optical particle counter data in the data files, minimizing file size while providing adequate resolution for optical calculations. Note that 20 bins/decade is narrow: each bin is only a 12% change in diameter. When establishing the methodology to derive concentrations from PALMS, F19 found that grouping the size-resolved PALMS composition data into just four size bins was a good compromise between adequate statistics and changes in particle composition with size. This choice of bin width is extensively and quantitatively evaluated in F19, which states "It is infeasible to retain the raw size resolution of the OPS [optical particle spectrometer, AMP here] for the integrated concentration analysis.

We respectfully disagree that there is only one proper way to investigate the size range from a complex instrument such as PALMS under the ATom conditions (i.e., the specific method used by Froyd et al. (2019), hereafter referred to as F19). Murphy et al. defend the method used to report concentrations in F19 as appropriate. We have not argued that the reported PALMS data products should be changed or processed differently. Instead, we are analyzing the data from multiple instruments towards a different goal, which is to compare the operational size coverage of these instruments. For this goal, analyzing the data at a higher resolution is useful. Also, for a paper comparing multiple instruments such as this one, an important criterion is to be able to represent all the instruments in the same way, which is not possible with the F19 method.

F19 assumes homogeneous particle composition within a bin, and sometimes extends composition information from part of one bin, where particles are detected, to the edges of that bin, where no particles were detected. This is especially the case at the edges of the smallest or the largest bin, as shown in Fig. S13 of the AMTD version, which was provided by the PALMS team. The assumption that the (undetected) particles at the edges of the bins have the same composition as those detected may be true or not depending on the ambient particle distribution sampled. The figure below shows a case study of size and composition evolution due to nucleation and condensational growth in the atmosphere (Zhang et al., 2004), conceptually similar to the growth of new particles into the Aitken mode in the upper troposphere observed during ATom (Williamson et al., 2019). The entire size range shown below is represented with only 3 bins in the PALMS archived data, but clearly there is evolution and differences across smaller size intervals in the study below. For this reason, it is perfectly legitimate to investigate the size coverage of the different instruments at a higher resolution than the choice of F19.

[Figure]

**FIGURE 3.** Time series of the mass concentrations of particle species and apparent volume (a) and evolutions of the size distributions for apparent volume (b), total mass concentration (c), and mass concentrations of chemical species (d−g) on September 12, 2002. The four stages (I−IV) of the nucleation and growth are marked on plot a. To the right of the corresponding image plot are the average size distributions of given parameter during these four stages (b′−g′). Particle apparent volume was calculated using SMPS number distribution data assuming spherical particles. Missing data (white areas in plot a and gray areas in plots b−g) are due to either occasional instrumental malfunction or maintenance/calibration. White areas in plots b−g are due to the omission of data points that are below the detection limit (1σ) of the AMS.

Fig. VIII (Figure 3 in Zhang et al. (2004)).

As shown in response to comment R1.2, the choice of 5, 10, or 20 bins per decade results in small differences, and any of them can be justified for our analysis. We prefer to illustrate the results with higher size resolution, also for the reasons discussed in response to R1.2.

Importantly, it is not correct to say that we are imposing more strict statistical criteria on the PALMS due to the higher bin resolution. We adjusted the criterion for deciding how many particles are needed in a bin to fully characterize the particle chemical composition to be consistent with F19. While F19 states that it requires $\geq 5$ spectra in each size bin that contribute significantly to the total volume (4 bins in total), we assume that if PALMS detects one particle in a given AMP size bin (at 20 bins/decade), the composition of the bin is fully characterized. The two criteria are roughly equivalent: the 4 bins with $\geq 5$ spectra means in total 20 spectra, while the AMP size resolution based analysis means in total 35 spectra (for the 35 size bins). The chosen criterion in this study is simply adapted from F19 to analyze the operation size range of PALMS during ATom studies. In reality, more particles might be required to fully characterize the particle composition for a narrow size bin at the higher size resolution for the possible existence of externally mixed aerosols or the variations of internally mixed aerosols. However, this conservative scaling implicitly acknowledges some degree of correlation between the higher-resolution bins.

To illustrate the effect of size resolution and time interval on PALMS detected particles, we included Fig. IX (below) in SI (slightly revised from Fig. S14 of the AMTD version). Again, as shown in this figure, the broad 4 bins smooth out the sharp decrease of particle counts on both edges compared to the other cases that utilize higher size resolutions. Even at 5 or 10 bins/decade, the edge effects are clearly visible. We prefer to use the data at 20 bins/decade to more clearly illustrate the effects and more easily compare with other instruments.

[Figure]

**Fig. IX (Fig. S15 in SI). (Top) PALMS detected particle numbers per size bin at several size resolutions at a 3-min time interval (i.e., the time resolution that the public PALMS-AMP mass products are reported at) and longer averaging time scales for ATom-2. (Bottom) The ATom-2 campaign averaged AMP number size distributions with the fraction that is characterized by PALMS-AMP.**

**S1.4.** The G20 manuscript creates a data coverage metric called "chemical information content" or "chemical composition information". The metric, used in G20 to derive Figs. 8, S29, and elsewhere, assigns an information content of 1 to size bins with particles and 0 to size bins without particles. It is

easy to see that this leads to lower values of "chemical information" as the number of size bins is increased. Imagine, for example, that an instrument has measured 50 particles. If there is no size resolution (one bin), the "chemical information" is always 1. If there are three bins and 50 particles, statistically each bin will almost always contain a particle and the average "chemical information" will be very close to one. But if there are 100 size bins (e.g. the native resolution of the UHSAS optical spectrometer) and 50 particles, at least 50 bins are necessarily empty and the average "information content" will be less than 0.5. With 1000 bins, in this example the "chemical information" metric that is supposed to quantify data coverage would be reduced to <0.05. Although of course G20 are not proposing 1000 bins, any metric that can produce arbitrarily small values of the supposed data coverage simply by choosing narrower bins must be mathematically invalid.

This is a misunderstanding, which becomes a straw man's argument, as discussed in response to comment S1.1. G20 does not define any metric called "chemical information content." As stated in the abstract, "The particle size ranges that contribute chemical composition information to the AMS and complementary composition instruments are investigated, to inform their use in future studies." A particle size range informs us of which particle sizes are contributing to the measurements of each one of the instruments. A particle size range is not a "metric of chemical information content."

As discussed above in S1.3, the analysis can be done with many reasonable size resolutions, which are relevant to real variations of composition in the atmosphere. And the criterion for judging how many particles are needed in a smaller size bin to characterize particle composition in G20 is similar to that in F19. The impact of using a higher size resolution has already been considered and adjusted for.

**S1.5.** The first reason the "chemical information" metric in G20 is flawed is that it penalizes high-resolution data for empty bins but does not give credit for information obtained from narrower bins. For example, having two size bins instead of one adds information about possible differences in

composition between small and large particles; G20 assign no value, either conceptually or mathematically, to this added information. Second, the G20 analysis does not recognize autocorrelation. In reality, neighboring bins are not independent of each other, either in concentration or chemistry. This is particularly true at the high resolution used by G20. For example, if 200 nm particles are composed of 80% sulfate, it is very unlikely that 225 nm particles are composed of pure organics. If 200 nm particles truly had a completely different composition than 205 nm particles, and those again different than 210 nm particles, etc., PALMS would need to sample an enormous number of particles to fully characterize the size-dependent composition. If all the particles from 100 nm to 5 m in a given air mass were exactly the same, PALMS would only need to sample a few particles. Naturally, the amount of data required is in between these extremes. When analyzing PALMS data coverage, G20 do not consider that different size particles may have correlated compositions, yet that assumption is implicit in the interpretation of AMS and SAGA data. Without it one would have no idea, for example, if a 50:50 organic-sulfate mixture represented mixed particles, small pure organic particles and large pure sulfate particles, or vice versa. The AMS collection efficiency (bounce) correction, which is based on bulk composition, also assumes similar particle compositions at different sizes. For example, if 150 nm particles were ammonium sulfate and 250 nm particles were sulfuric acid, the AMS bounce correction would be incorrect.

As discussed in response to comments S1.3 and S1.4 above, the higher size resolution analysis presented in this study is just a more realistic illustration of the PALMS-AMP size coverage based on the broad 4 bins analysis in F19 without assuming homogeneous particle composition within a size bin and 100% size coverage from the reported size range, 100-5000 nm. Indeed, the adjacent size bins typically have similar or correlated compositions but that doesn't affect the analysis in this study since we adjusted the statistical criterion for the effect of choosing a smaller size bin, as discussed in response to comment S1.3. As we showed in response to that comment, similar results are obtained with 5 or 10 bins/decade, and real variations in composition can be found in the atmosphere over bins of those sizes.

SAGA is a bulk measurement of aerosol composition while AMS provides both bulk and size-resolved characterization in two operation modes. The size-resolved aerosol composition provided by PALMS-AMP is complementary to those from SAGA and AMS, and acknowledged in the text. Each instrument has a different size range over which it provides composition data. The overlap and differences are often confusing to users of the data, and one intent of this paper is to illustrate those facts in an easy-to-understand way. The strong opposition by the PALMS team to this simple characterization is perplexing. To make the bulk vs. size-resolved composition measurements clearer, we have added the following text to the paper:

"**It should be noted that the PALMS-AMP characterizes size-resolved rather than bulk aerosol composition, such as SAGA filter or AMS FMS data products (the AMS ePToF mode does measure size-resolved aerosol composition).**"

As stated above, there is no "chemical information metric" in G20. Fig. 8 in G20 just shows in easily digestible form what particles each instrument actually samples for a given slice of the ATom domain. This is completely independent of how each instrument processes and reports that data. We are characterizing here the particle size ranges that are analyzed by each measurement, independent of the idiosyncrasies of the chemical detection of each instrument (that are already well-known, and the subject of many publications). Otherwise, the comparison of the size ranges would become so ambiguous and complex as to be unfeasible.

While it is certainly true, as pointed out by the PALMS team, that AMS $CE$ can be impacted by externally mixed aerosol and can therefore cause a quantification bias, we fail to see how this is relevant to the discussion of this particular paper. To state this again as simply as possible: AMS $CE$ is an uncertain parameter whose impact on quantification uncertainty needs to be evaluated for data products

(which are NOT the subject of Fig. 8 in G20). However, the *CE* during ATom was very high (~0.9 on average, as documented on page 8 and shown as the 2D plot of *CE* vs. altitude in Fig. S10 of the AMTD version) and thus contributes less uncertainty than usual. We have evaluated the uncertainty separately in this paper by conducting extensive volume intercomparisons. As Fig. 3 shows fairly conclusively, for ATom at least, *CE* calculated with the mixed aerosol assumptions was able to reproduce the AMP volume within uncertainties, making it highly unlikely that large deviations in estimated vs. true *CE* impact ATom quantification.

For the PALMS case, Fig. 8 in G20 shows that its size range is significantly different from those of the other instruments, and documents that in graphical form. For certain conditions, the edges of the smallest and largest bin did not record many particles for short time-averages. If there is a composition gradient across those areas, there will be a bias in the PALMS-AMP products. These biases could be insignificant in many parts of the atmosphere but significant in others.

We believe it is useful to inform the scientific community more clearly about these instrument-to-instrument differences. That is the value of showing the size coverage as in Fig. 8 in G20. F19 certainly discusses these biases, but in our experience (e.g., working with many aerosol modelers), those highly technical discussions are too difficult to translate into how model evaluation should be conducted. On the other hand, we find in our interactions that Fig. 8 in G20 is immediately understood, giving modelers a useful guide about how to use the different ATom instruments to investigate the chemical composition of the particles *of their interest*. As will be shown below in response to comment S1.10, our results are fully quantitatively consistent with those from the PALMS team. It is useful when different teams represent the same complex scientific information in different ways for different purposes.

**S1.6.** A rigorous analysis of information and data coverage is beyond the scope of this comment but we can point out that that it should be framed in terms of detection limits. The G20 "chemical information" metric conflates information about the instrument with information about the atmosphere. In a typical 3-minute period in the stratosphere PALMS measures no 30 nm particles and no 3 μm particles. G20 assign the same zero value to these, yet they are very different. The former is a statement about PALMS because 30 nm is far below PALMS' nominal size range, whereas the latter is a statement about low concentrations in the stratosphere. In G20 this is evident in Fig. 8 for example, where zero "information content" is assigned at high altitudes to large sizes that PALMS samples well (because the atmosphere there contained almost no large particles). This illustrates why the scientific literature conventionally discusses detection limits for in-situ instruments rather than "chemical information". When one compares an instrument detection limit with an atmospheric concentration it properly separates instrument performance from atmospheric properties. There are mathematically rigorous ways of defining information content for aerosol size distributions (Preining, 1972) in the context of a priori information (Petty, 2018), yet an arbitrary metric is developed in G20 instead.

As discussed above in response to comments S1.1, S1.4, we did not define any metric of "chemical information." That is simply a misunderstanding, that becomes a repeated "straw man's argument" in this short comment. We are aware of information theory, Shannon's entropy, and similar concepts, but those concepts are not relevant to the current manuscript.

We are trying to represent a simpler concept, namely which particle size ranges contribute and not to the measurements of each instrument under ATom conditions. Like the field campaign and the data, this analysis must include information about both the instrument and the atmosphere. This is a natural feature of the analysis, and it is not a problem in any way.

DLs are in general irrelevant here, since they concern the features of the chemical detector within each instrument, not what particles are sampled. Again, we aim to show which particle sizes *are and are not* analyzed by each instrument — not what *could* be measured *if* the ambient concentrations were different. That is a specific analysis, that we believe to be of high interest based on our experience. It is a different analysis than what was carried out in F19, and naturally it uses slightly different methods and graphical representations. One could choose to conduct various other analyses, such as the ones suggested by the PALMS team about trying to quantify information in "information units", or analyzing the detection limits, or the myriad other possibilities when dealing with such a collection of very complex instruments. Other scientists are welcome to perform such analyses, but in our experience, the particular analysis of the size coverage in this paper is especially important. This is because we have encountered much confusion about this topic in our interactions with the scientific community during the last 20 years. There is no reason why only one type of analysis should be allowed for a specific instrument, or why rigid control about which analyses are permissible should be given to the instrument PIs. We believe our analysis to be correct and useful, and it is quantitatively consistent with the PALMS team's own analysis.

Our method of estimating the PALMS detected particles is straightforward by applying the PALMS detection efficiency curve to the AMP measured particle numbers and scaled to the reported PALMS positive mode spectra to ensure a proper evaluation. And the estimation based on the publicly available data is consistent with the limited data shared by the PALMS team in their comment (see details below in S1.10). We have added the following text to SI to further clarify this point:

"**The PALMS detected particle curve appears to be an inverted U-shape. For the small particle end (< 200 nm in Fig. S14), the very sharp decrease in the PALMS detection efficiency curve (Fig. 6 in Froyd et al.** (2019)**) dominates over the increasing atmospheric particle concentrations at smaller sizes reported by AMP. For the large particle end (> 2.5 µm), fewer large particles were**

**present in the air despite the slight increase in the PALMS detection efficiency curve at larger sizes.**"

**S1.7.** The G20 manuscript applies different metrics to different instruments, leading to biased comparisons. In particular, G20 Fig. 8, the primary comparison of instrument measurement capabilities, is not internally consistent. The "chemical information" metric is applied only to PALMS, not the other instruments. PALMS data coverage is derived using a limited (3 minute) sample time. Imposing this sample time restriction only on PALMS misrepresents its size range and data coverage relative to the other instruments. Were a similar information content or sample time restriction imposed on AMS and SAGA, a high fraction of samples would be below detection limit. At the native sampling times for AMS or SAGA (1 min and 5-15 min, respectively), 72% of AMS and 35% of SAGA samples are below their detection limits for a major chemical component (sulfate, ammonium, nitrate, or organic material). Such samples have little "chemical information" across all sizes beyond being able to say "below detection limit", yet that is ignored in G20 Fig 8, where instead AMS and SAGA are shown to detect 100% of particles within their nominal size ranges. On the same 3 minute sample time as PALMS, over 60% of AMS organic data in the tropical Pacific are below detection limit, yet G20 assert that those data have more "chemical information" than PALMS data above detection limit.

See response to comment S1.6.

To re-state, we make no statement on how well the instruments detect the composition of those particles, only on if they sample those particles. We stated that Fig. 8 in G20 illustrates the approximate particle size ranges being sampled (without consideration of the details of the chemical detection), which is consistent for all instruments, including PALMS. Specifically, our analysis does not consider how efficiently PALMS detects and quantifies the composition of each ablated particle. The different methodology required in characterizing the PALMS-AMP product size range is due to the fact that the

PALMS is a single-particle technique, different from a bulk measurement. To ensure a proper evaluation of the PALMS-AMP, the criterion of judging counting statistics is equivalent to F19, despite choosing a different size resolution, as discussed in response to comment S1.3 above.

And while detection limits are irrelevant for our specific analysis, we would like to point out that the logic in this comment S1.7 is flawed: e.g., both SAGA and AMS report particulate nitrate as a standard aerosol species. And in most of the ATom domain, the aerosol is acidic enough that all inorganic nitrate resides in the gas phase (Guo et al., 2016; Nault et al., 2021). Hence both instruments report nitrate below DL in all of these cases, which is exactly what should happen and has no bearing on their overall ability to characterize the full aerosol composition, as the volume comparisons in this paper clearly show, even if some of the AMS species are under the specific DL. In other words, the fact that a species is under DL is very useful information. If nitrate was not measured by any instrument, we would not know anything about its presence or absence from different locations in the atmosphere. If we know it is below a certain DL within a particle size range, that is very useful information. And even at a very small total volume concentration, such as 1 $nm^3$ $scm^3$, the SAGA and AMS instruments are still providing this useful information over the full size range that they can sample.

The 3 min sample time interval is how the PALMS-AMP data are archived, thus our first choice for the base case. Three minutes correspond to ~1.5 km ascent/descent and ~36 km horizontal distance in the atmosphere under the typical ATom flight profile, which is relevant to the analysis of many events. Many plumes and air masses in ATom have those or smaller scales. Since averaging over a longer time increases the size range contributing information to the PALMS measurements, we also presented the wider observable size ranges of PALMS-AMP if a 60 min time interval is chosen or the F19 method that assumes 100% size coverage from 100 nm to 5000 nm in Fig. 8 (G20) and Fig. S29-30 (G20 SI). We note that the fact that averaging time does not affect the size ranges sampled by SAGA and AMS, while it does affect the PALMS range, is just a feature arising from the very different measurement

methods. We view documenting this type of feature as highly useful for users of the data, and fail to see why this would be scientifically incorrect.

**S1.8.** A quantitative error in G20 is that their derived PALMS+AMP concentrations in Fig. 8 and elsewhere scale improperly with sampling time. This is illustrated in Fig. S29, where increasing the PALMS sampling time from 3 to 60 min increases the PALMS+AMP number and volume. Although PALMS observes more particles with longer sample time, higher particle counts do not translate into higher derived concentrations. The G20 method to derive PALMS+AMP concentrations is therefore flawed: physical concentrations do not scale with sample time. The method of F19 does not have this problem.

Fig. 8 in G20 makes no statement about data products, and from going over the past correspondence with the PALMS team this seems to be the most important misunderstanding. We are not proposing that the data products be changed, or that the current data products are not valid. We are merely trying to inform data users and the scientific community about real differences in the size coverage from different instruments, and most importantly, where they matter and where they can be neglected.

Our paper never states that the PALMS-AMP reports higher concentrations with longer averaging time, which would obviously be incorrect. As the manuscript makes clear, the PALMS mass concentration product is based on the particle volume above 100 nm. We do note that F19 actually stated 60 nm, but the PALMS team has revised this limit since then. What Fig. 8 in G20 does show is that for short periods (the reported 3 min time interval) in most of ATom the PALMS will not see a single particle below 130 nm. As shown in Fig. S13 (provided by the PALMS team), the composition of 100 nm particles will mostly be assigned based on that measured at 200 nm. This is a useful approximation and may work in many situations, but not in others, depending on the composition gradients in the atmosphere. We intend to inform data users about these details for all instruments analyzed (not just

PALMS), so that they can for example better interpret data vs. model comparisons for phenomena of their specific interest.

**S1.9.** Figure 1 below shows that it is essential to use information across wider bins than 20/decade to obtain accurate concentration measurements with PALMS+AMP. If one simply puts zeros in empty bins, as in G20 Fig. 8, the resulting concentrations have large discrepancies and are biased low.

As stated above in response to comment S1.8, G20 does not suggest that the PALMS data products need to be modified. This is just a misunderstanding. Our analysis addressed a different point, namely showing which particle size ranges are characterized by the different instruments.

**S1.10.** The G20 manuscript also strongly underestimates the number of particles sampled by PALMS. Fig. S14 in G20 is the basis for the derived PALMS+AMP concentrations in Fig 8 and elsewhere. Although their estimated sampling rate near the peak sampling efficiency at around 300 nm appears reasonable, their estimates have large errors for smaller and larger particle sizes, exactly those most important for data coverage (Figure 2). For instance, for the narrow bin at 1005-1128 nm G20 calculate that PALMS observed 500 particles over an ATom deployment, whereas PALMS actually observed over 14000. Considering the 5 bins/decade curve on G20 Fig. S14, they underestimate the average sampling rate for the bin centered near 130 nm by a factor of about 10 and underestimate the PALMS sampling rate above 1 m by factors of 8 to 40. Similar errors apply to all other curves in S14. These errors presumably propagate into all calculations in the manuscript (Fig 8, 9, S15, S28, S29, S30, and Table S2). Underestimating the number of particles sampled by PALMS by x10 and higher factors will seriously affect the conclusions of the manuscript.

First of all, we want to clarify that in principle, there is no need for us to have to estimate the number of particles sampled by PALMS. This quantity is part of the PALMS dataset and easily available to the

PALMS team. We have requested this information many times over more than a year, but the PALMS team has refused to share it. While this is certainly not a primary, archivable data product, it is relevant level 1 ancillary instrument data that in our view falls under the NASA open data policy and should be made available to third parties regardless of specific purpose. But nevertheless, it is only this refusal that forced us to estimate the number of particles counted. We would be very happy to revise Fig. 8 (G20) using the *actual* particle numbers if the PALMS team would provide these data. So it is curious, to say the least, that the PALMS team complains about an error that they refuse to help remove.

Secondly, we strongly disagree that there are "large errors" in the PALMS detected particle numbers between this study and Fig. 2 posted by the PALMS team. That figure is not an apples-to-apples comparison of the two quantities, for two reasons:

(1) The PALMS team plotted total (positive+negative) spectra, while we only plotted positive spectra which are used to derive the main PALMS-AMP mass products, such as OA and $SO_4$, according to F19. Importantly, PALMS utilizes one polarity at a time, hence the sum of both modes is not really relevant, since only one set of particle types is detected at a given time.

(2) Fig. S14 of G20 only shows the ATom-2 campaign, which was the campaign that had the lowest particle concentrations across all 4 sub-campaigns. Fig. 2 of the short comment, rather than comparing only ATom-2 as would be expected, shows all 4 campaigns but emphasizes the average with the black line.

We do agree that our estimation is not perfect, since there are inevitable uncertainties when we are forced to reconstruct the particle counts from scratch based only on the publicly available data. Our estimation is based on the reported ATom-1 detection efficiency curve for the free troposphere in F19 (since other curves have not been made public or provided to us). Thus it is understandable that the actual counts may be slightly different and the other AToms may not follow the curve perfectly, but it is

the best that could be done at the time of submission with the information made available. As shown in Fig. 6 of F19, the PALMS detection efficiency can change with altitude, particle composition, inlet performance, and unknown factors between different cases. Additionally, the effects of software or hardware limitations were not considered in the average PALMS detection efficiency curve in ATom-1.

We have replotted the data as in Fig. 2 of the short comment below, using the average of all ATom campaigns. On the left panel, we can see that our estimate of the number of particles counted is very close to the actual number, except for the left-most bin, where the particle counts are underestimated by a factor of 4.5. Now that (finally) some data on actual particle counts have been provided in Fig. 2 of the short comment, we can finally perform a small correction to remove this disagreement. We moved the detection efficiency curve by -10 nm for Bin 1 (100-240 nm), a correction which is within the reported uncertainty in Fig. 6 of F19. This brings the estimate of the particles counted by our method at this bin to a reasonable agreement (a factor of 2). We have propagated this correction for all the graphs concerning PALMS in the revised manuscript.

The 24x difference in the narrow bin at 1005-1128 nm, 14000 vs 581 (581 is the precise number in the figure) as pointed out by the PALMS team, is actually only a factor of 1.6 difference as shown in Fig. X (Fig. S14 in SI) below for the reasons #1 and #2 above. Also, it is unclear where the 14000 number comes from, and to which specific deployment it applies. In addition, we discovered an error in our data processing for that figure only. We updated Fig. IX (Fig. S15) for a proper campaign average by averaging the data points directly, which increased the estimation of particle counts by ~3 times, instead of scaling vs. the 3-min case based on the ratio of averaging time intervals. This doesn't affect any discussion since the ATom-2 campaign average curve in that figure was merely for illustration, and was not used anywhere else in the paper. Despite the minor differences, the plot made by the PALMS team also illustrates that PALMS detected far fewer particles at both ends than the mid-size range, which is consistent with Fig. 8 in G20. A revised version of Fig. 8 after propagating this modification is shown

below. The minor impacts on Fig. 8 and Fig. 9 are shown below by comparing the AMTD and the updated versions. Hence, we conclude that these minor differences are expected, given that the actual count data have not been provided, and do not impact the validity of the results shown in Fig. 8 and elsewhere in the manuscript.

[Figure]

Fig. X (Fig. S14 in SI): **(a) Comparison of the PALMS detected particle numbers over 3 mins time interval between Murphy et al.** (2020) **and the estimates in this study based on the original ATom-1 PALMS detection efficiency curve in Fig 6 of Froyd et al.,** (2019)**. The data are shown at a resolution of 5 bins/decade, for consistency with Murphy et al.** (2020)**. The bottom panel shows the ratio between the two methods. (b) Sensitivity tests by moving the PALMS detection efficiency 10 nm smaller for Bin 1 (100-240 nm; thus leading to higher particle counts in this size range), all else stays the same. The 10 nm is roughly the reported uncertainty of the PALMS detection efficiency curve, while the actual uncertainty is probably larger when considering the limitations in software or hardware.**

[Figure]

Fig. XI. Comparison of Fig. 8 and Fig. 9 in the main text between the AMTD and the updated versions.

**S1.11.** Specifically, G20 state (line 345) "The probability of detecting on average one valid particle per AMP size bin in the PALMS is very low below 160 nm and above 1000 nm over a typical 3 min analysis period." In contrast, on average PALMS actually measured in 3 minutes the composition of about 4 particles smaller than 160 nm and about 10 particles larger than 1 m. G20 (lines 336 and 694) also refer to PALMS+AMP products for small and large particles as "extrapolations". In reality, during the ATom deployments PALMS measured over 100,000 particles between 100 and 180 nm diameter and almost 100,000 between 1 and 4 μm.

As discussed in detail in the response to comment S1.10, the estimated PALMS detected particles in this study is in good agreement with the curve posted by the PALMS team. Importantly, campaign averages such as the ones cited by the PALMS team (averaging over 600 hrs of flight data), while certainly useful, tend to obscure in which parts of the atmosphere these particles were actually measured, which is the point of our analysis. Many more particles can be detected in the boundary layer, biomass burning plumes, and other areas of higher concentrations. However, the focus of our analysis is the average conditions at each altitude bin. Instead of the overall numbers as given in this comment, making the list of particles counted (detection time and size) public would allow other researchers to investigate these issues as needed for future analyses.

**S1.12.** We provided the AMS team with unpublished data on the PALMS sampling rate (the red curve on their Fig. S13), and allowed them to use these data even after we were removed as coauthors. We have been unable to replicate exactly how they arrived at their underestimates of PALMS sampling rates. The manuscript states (Fig. S14 caption) that they used a detection efficiency and multiplied it by a flow rate and atmospheric concentrations. However, according to F19, using a detection efficiency curve for PALMS is "not recommended due to many possible pitfalls and large, unquantifiable errors.

We appreciate that the PALMS team shared the PALMS relative data coverage within each size bin. Regarding their criticism: as noted now multiple times, our analysis is completely agnostic on how the convolution of the AMP and single-particle data is done. That all happens "downstream" of Fig 8. We understand that some single-particle MS groups have scaled their mass products by the detection efficiency and that F19 takes a different approach. But that has nothing to do with which particle sizes range PALMS is actually sampling at each size range and altitude, which is the point of Fig. 8 in G20.

**S1.13.** Summary. The incorrect calculation of the number of particles sampled by PALMS and the use of arbitrarily narrow bin widths together lead to low assessments of the PALMS data coverage. G20 assert (line 691, also Table S2 and Figs 8 and 9) that the PALM+AMP data characterize about 54% of the aerosol volume within 3 minutes of sample time during the ATom flights. Figure 3 shows cumulative size distributions for the ATom flights. One can see that a 54% fraction of the volume within the nominal PALMS size range is implausibly low.

The text in comment S1.13 is the 1$^{st}$ paragraph of the summary from the PALMS team. The issue of bin width has been addressed in response to comments R1.2 and S1.3, while the good agreement of our particle count estimates with the averages provided by PALMS has been addressed in response to comment S1.10. And the seemingly low 54% fraction (now 56%) of the AMP volume observed by the nominal PALMS size range has been explained in response to comment R1.4 above.

**S1.14.** The analysis of the PALMS+AMP data in G20 has both quantitative and conceptual errors. Even if the quantitative errors were fixed, the "chemical information" metric of data coverage would still be conceptually flawed. G20 Figs. 8 and 9 and the stated size-dependent numbers of PALMS-sampled particles are incorrect by large margins, as is the 54% PALMS volume coverage figure. Finally, instruments are not compared using the same criteria: only the PALMS data are scaled by "chemical

information" using a limited sample time, and an implicit assumption that particle composition is uncorrelated across nearby diameters is applied only to PALMS.

All the points have been answered above, and they all appear to be due to misunderstandings of our analysis:

- See response to S1.10 for the quantitative agreement between our analysis and the PALMS data, after a small correction is implemented using the newly provided data.
- See response to S1.1 and S1.4 for the discussion of the "chemical information metric" being a straw-man argument.
- See response to S.1.7 for the consistency of the treatment of the different instruments, given the goals of our analysis.

We hope that this response helps resolve most of these misunderstandings and to clarify our analyses.

**S1.15.** The manuscript is not suitable for publication unless the incorrect calculations, the "chemical information" analysis, and all associated discussion are removed. Specifically, in G20, Figs. 8, 9, S14, S15, S28, S29, and S30 show incorrect information about PALMS. The associated discussion starting on line 331 is incorrect, as are the data coverage percentages following line 689, line 744, and in Table S2. The red curve on Fig. S13 is correct but mislabeled.

As summarized in response to comment S1.14, this criticism turns out to be incorrect and probably caused by misinterpretation of the independent analysis of PALMS in this study, the request of removing this realistic analysis is not justified. Moreover, any small remaining quantitative differences are just due to the PALMS team refusing to provide the requested data over a period of a year. With the requested data, any such differences could have been completely removed from our analyses. Although our analysis is constructed based on the limited and publicly available data, as shown in Fig. X in

response to comment S1.10, the deviation or uncertainty in the estimated particle numbers are within a factor of 2 to that posted by the PALMS team. This minor difference doesn't significantly affect the conclusion of this analysis, i.e., the operational size coverage of PALMS during ATom.

Despite the lack of agreement on this paper with the PALMS team, we look forward to collaborating with them in the future.